# Bellman Diffusion: Generative Modeling as Learning a Linear Operator in the Distribution Space

## Abstract

Deep Generative Models (DGMs), including Score-based Generative Models, have made significant progress in approximating complex continuous distributions. However, their application to Markov Decision Processes (MDPs), particularly in distributional Reinforcement Learning (RL), is underexplored. The field remains dominated by classical histogram-based methods, which suffer from discretization errors, leading to instability and slower convergence. This work highlights that this gap stems from the nonlinear operators used in modern DGM's modelings, which map neural network functions to the target distribution. These nonlinearities conflict with the linearity required by the Bellman equation, which relates the return distribution of a state to a linear combination of future states' return distributions. To address this, we introduce *Bellman Diffusion*, a new DGM that preserves the necessary linearity by modeling both the gradient and scalar fields. We propose a novel divergence-based training technique to optimize neural network proxies and introduce a new stochastic differential equation for sampling. With these innovations, Bellman Diffusion is guaranteed to converge to the target distribution. Our experiments show that Bellman Diffusion not only achieves accurate field estimations and serves as an effective image generator, but also converges $1.5\times$ faster than traditional histogram-based baselines in distributional RL tasks. This work paves the way for the effective integration of DGMs into MDP applications, enabling more advanced decision-making frameworks.

## 1 Introduction

Markov Decision Processes (MDPs), particularly distributional Reinforcement Learning (RL) (Bellemare et al., 2017), learn the *distribution* of returns rather than just the expected value (i.e., the Q-function). This allows the model to capture the intrinsic randomness (stochastic dynamics and rewards) of returns, demonstrating its efficacy and broad applicability (Lowet et al., 2020; Lyle et al., 2019; Dabney et al., 2018). Despite Deep Generative Models (DGMs), such as Energy-Based Models (EBMs) (Teh et al., 2003), Generative Adversarial Networks (GANs) (Goodfellow et al., 2020), and emerging Score-based Generative Models (SGMs) (Sohl-Dickstein et al., 2015; Song et al., 2021; Ho et al., 2020), are well-developed for learning complex continuous distributions, their application to Markov Decision Processes and distributional RL remains underexplored[1]. Instead, classical histogram-based methods (e.g., C51 (Bellemare et al., 2017)) remain widely used in MDPs: These methods leverage Bellman's equation (Bellman, 1954; Mnih et al., 2013) to efficiently update the model with partial trajectories and approximate return distributions using discrete bins, rather than directly modeling the continuous return distribution. The discretization inherent in these methods can accumulate errors, causing instability and slower convergence. Specifically, for continuous return distributions, histogram-based methods inevitably introduce discretization errors at each state, leading to error propagation along the state-action trajectory. This accumulation of errors, combined with potentially long trajectories, makes histogram-based methods, unstable to

---

[1]A naive approach to modeling the return distribution with DGMs is to sample full state-action trajectories and use the computed returns to train EBMs. However, this method is not scalable, as trajectory sampling is costly in many RL environments (see Appendix D for more details).

train and difficult to converge. This highlights the need to leverage a continuous DGM to model continuous return distributions in the Bellman update.

In this work, we first identify the main reason for the gap in applying modern DGMs to MDPs: the *linear nature of the Bellman equation update*. More precisely, Bellman equation relates the return distribution $p_z$ of a state $z$ as a *linear combination* of return distributions $p_{z'}$ of future states $z'$, expressed formally as:

$$p_z(\cdot) = \sum_{z',r} \alpha_{z,z',r} p_{z'}\left(\frac{\cdot - r}{\gamma}\right), \tag{1}$$

where "$\cdot$" indicates a dummy argument, $r$ is the expected reward for transitioning between states, and $\alpha_{z,z',r}$, $\gamma$ are constants determined by the RL environment. However, the modeling operators of modern DGMs, which map neural network functions to target densities or related statistics, are inherently nonlinear with respect to the neural network functions themselves (note that the nonlinearity is not referred to the input data). This nonlinearity conflicts with the Bellman update's linear structure, which relates a state's return distribution linearly to those of future states, fundamentally hindering the direct application of existing DGMs in Bellman updates. Below, we use EBMs, effective in distribution approximation (Lee et al., 2023), as an example to further illustrate this point.

**Illustrative example.** At each state $z$, the EBM's modeling operator $\mathcal{M}_{\text{EBM}}$ maps the neural network function $E_z$ (known as energy) to the target (return) distribution $p_z$. This mapping is formally expressed as: $\mathcal{M}_{\text{EBM}}\colon E_z(\cdot) \mapsto \frac{e^{-E_z(\cdot)}}{Z_z} \approx p_z(\cdot)$. where $Z_z := \int e^{-E_z(\mathbf{x})} \, d\mathbf{x}$ is the normalization factor at each state $z$. In general, $\mathcal{M}_{\text{EBM}}$ is a nonlinear operator with respect to the input function $E_z$, which means the Bellman equation update cannot be used to link future state densities with the current state for efficient updates:

$$p_z(\cdot) = \mathcal{M}_{\text{EBM}}(E_z)(\cdot) \neq \sum_{z',r} \alpha_{z,z',r} \mathcal{M}_{\text{EBM}}(E_{z'})\left(\frac{\cdot - r}{\gamma}\right) = \sum_{z',r} \alpha_{z,z',r} p_{z'}\left(\frac{\cdot - r}{\gamma}\right).$$

As such, $\mathcal{M}_{\text{EBM}}$, acting as a nonlinear operator, disrupts the linearity of the distributional Bellman equation and rendering EBMs inapplicable in this context. In Sec. 2, we analyze the modeling approaches of other modern DGMs and find that none can preserve the linearity of the Bellman update, limiting the application of powerful DGMs in MDP tasks.

**Our framework: Bellman Diffusion.** To address this bottleneck, we propose *Bellman Diffusion*, a novel DGM designed to overcome bottlenecks in applying DGMs to MDPs. The core idea is to model the gradient field $\nabla p_z(\cdot)$ and scalar field $p_z(\cdot)$ directly. That is, $\mathcal{M}_{\text{Bellman}}\colon p_z(\cdot) \mapsto \begin{bmatrix} \nabla p_z(\cdot) \\ p_z(\cdot) \end{bmatrix}$. Since the gradient and identity operations are linear operators, the linearity of the Bellman equation is well preserved under $\mathcal{M}_{\text{Bellman}}$. For instance, after applying the gradient operator $\nabla$, the Bellman equation still holds:

$$\nabla p_z(\cdot) = \sum_{z',r} \frac{\alpha_{z,z',r}}{\gamma} \nabla p_{z'}\left(\frac{\cdot - r}{\gamma}\right).$$

We now use $p_{\text{target}}$ to denote the target density of each state, replacing the previous notation $p_z$. Since $\nabla p_{\text{target}}(\cdot)$ and $p_{\text{target}}(\cdot)$ are generally inaccessible, we introduce field-based divergence measures and transform them into feasible training objectives: approximating fields $\nabla p_{\text{target}}(\cdot)$ and $p_{\text{target}}(\cdot)$ with neural network proxies $\mathbf{g}_\phi$ and $s_\varphi$.

Given these proxies, we introduce a new sampling method: Bellman Diffusion Dynamics, associated with the fields represented by the following stochastic differential equation (SDE):

$$d\mathbf{x}(t) = \underbrace{\nabla p_{\text{target}}(\mathbf{x}(t))}_{\approx \mathbf{g}_\phi} dt + \underbrace{\sqrt{p_{\text{target}}(\mathbf{x}(t))}}_{\approx \sqrt{s_\varphi}} d\mathbf{w}(t), \quad \text{starting from } \mathbf{x}(0) \sim p_0, \tag{2}$$

where $\mathbf{w}(t)$ is a Brownian process and $p_0$ is any initial distribution. Once the fields are well approximated, we can replace the field terms in the above equation with learned proxies, resulting in a proxy SDE that can be solved forward in time to sample from $p_{\text{target}}(\mathbf{x})$.

**Theoretical and empirical results.** Theoretically, we guarantee the convergence of our Bellman Diffusion Dynamics to the stationary distribution $p_{\text{target}}$, regardless of the initial distribution (Theorem 4.1), and provide an error bound analysis accounting for neural network approximation errors (Theorem 4.2). Thus, Bellman Diffusion is a reliable standalone generative model.

Experimentally, we show the generative capabilities of Bellman Diffusion on real and synthetic datasets, confirming accurate field estimations, with promising results in image generation. We further apply Bellman Diffusion to classical distributional RL tasks, resulting in much more stable and $1.5\times$ faster convergence compared to the widely used histogram method. Notably, it can effectively learn and recover the target distributions with multiple unbalanced modes, a challenge for score-based methods (Song & Ermon, 2019) due to the inherent nature of the score function.

In summary, Bellman Diffusion introduced in this paper stands as a novel and mathematically grounded generative modeling approach, paving the way for continuous density modeling in various applications within MDPs, such as Planning and distributional RL.

## 2 LINEAR PROPERTY FOR MDPs

In this section, we review modern DGMs and highlight the desired property to facilitate density estimation with Bellman updates, avoiding full trajectory updates.

### 2.1 MODELINGS OF MODERN DEEP GENERATIVE MODELS

DGMs aim to model the complex target distribution $p_{\text{target}}$ using a neural network-approximated *continuous* density, enabling new samples generation. Below, we review well-known DGMs and offer high-level insights into how they define a *modeling operator* $\mathcal{M}$ that connects their own modeling functions to the desired density or its related statistics.

**Energy-Based Models (EBMs) (Teh et al., 2003):** These models define an energy function $E(\mathbf{x})$ and represent the probability as: $p_{\text{target}}(\mathbf{x}) \approx \frac{e^{-E(\mathbf{x})}}{Z}$, where $Z := \int e^{-E(\mathbf{x})} \, \mathrm{d}\mathbf{x}$ is the partition function for normalizing probabilities. EBM defines a modeling operator $\mathcal{M}_{\text{EBM}} \colon E(\cdot) \mapsto \frac{e^{-E(\cdot)}}{Z}$, linking the statistic $E(\cdot)$ to desired density.

**Flow-Based Models (Rezende & Mohamed, 2015; Chen et al., 2018):** These use a series of invertible transformations $\mathbf{f}(\mathbf{x})$ to map data $\mathbf{x}$ to a latent space $\mathbf{z}$, with an exact likelihood: $p_{\text{target}}(\mathbf{x}) \approx \pi(\mathbf{z}) \left| \det \frac{\partial \mathbf{f}^{-1}(\mathbf{x})}{\partial \mathbf{x}} \right|$. It determines a modeling operator $\mathcal{M}_{\text{Flow}} \colon \mathbf{f}(\cdot) \mapsto \pi(\mathbf{f}(\cdot)) \left| \det \frac{\partial \mathbf{f}^{-1}(\cdot)}{\partial \mathbf{x}} \right|$, connecting the transformation $\mathbf{f}(\cdot)$ to desired density.

**Implicit Latent Variable Models:** These models define a latent variable $\mathbf{z}$ and use a generative process $p(\mathbf{x}|\mathbf{z})$, where the latent space is sampled from a prior $\pi(\mathbf{z})$, usually taken as a standard normal distribution. Two popular models are VAE (Kingma, 2013) and GAN (Goodfellow et al., 2020). VAE maximizes a variational lower bound using an encoder network $q(\mathbf{z}|\mathbf{x})$ to approximate the posterior distribution, while GAN employs a discriminator to distinguish between real and generated data, with a generator learning to produce realistic samples but lacking an explicit likelihood. Since VAEs and GANs are implicit models, they lack an explicit modeling operator like $\mathcal{M}_{\text{EBM}}$ and $\mathcal{M}_{\text{Flow}}$ that connects modeling functions to the desired density or its related statistics.

**Score-Based Generative Models (SGMs) (Song et al., 2021):** They involve a process that gradually adds noise to $p_{\text{target}}$, resulting in a sequence of time-conditioned densities $\{p(\mathbf{x}_t, t)\}_{t \in [0,T]}$, where $t = 0$ corresponds to $p_{\text{target}}$ and $t = T$ corresponds to a simple prior distribution $\pi(\mathbf{z})$. Then, SGMs reverse this diffusion process for sampling by employing the time-conditioned score $\mathbf{s}(\cdot, t) := \nabla \log p(\cdot, t)$ and solving the ordinary differential equation (Song et al., 2021) from $t = T$ to $t = 0$ with $\boldsymbol{\phi}_T(\mathbf{x}_T) = \mathbf{x}_T \sim \pi$: $\mathrm{d}\boldsymbol{\Psi}_t(\mathbf{x}_T) = \left( \mathbf{f}(\boldsymbol{\Psi}_t(\mathbf{x}_T), t) - \frac{1}{2} g^2(t) \mathbf{s}(\boldsymbol{\Psi}_t(\mathbf{x}_T), t) \right) \mathrm{d}t$, where $\mathbf{f}$ and $g$ are pre-determined. This flow defines a pushforward map $\mathcal{V}^{T \to t}[\mathbf{s}]$ of the density as $\mathcal{V}^{T \to t}[\mathbf{s}]\{\pi\} := \pi \left( \boldsymbol{\Psi}_t^{-1}(\cdot) \right) \left| \det \frac{\partial \boldsymbol{\Psi}_t^{-1}(\cdot)}{\partial \mathbf{x}} \right|$. Thus, SGMs determine a modeling operator $\mathcal{M}_{\text{SGM}} \colon \mathbf{s} \mapsto \mathcal{V}^{T \to 0}[\mathbf{s}]\{\pi\} \approx p_{\text{target}}$.

## 2.2 Desired Linear Property in MDP

As the case of EBMs shown in Sec. 1, to leverage the strong capability of DGMs in density modeling with the Bellman update (Eq. (1)), the linearity of modeling operator $\mathcal{M}$ is crucial:

> ***Linear property of modeling.*** *The modeling operator $\mathcal{M}$ defined by a DGM is linear:* $\mathcal{M}(af + bg) = a\mathcal{M}(f) + b\mathcal{M}(g)$, *for any reals $a, b$ and functions $f, g$.*

If $\mathcal{M}\colon f_z(\cdot) \to p_z(\cdot)$ is linear, we can link future state densities or their statistics with the current state for efficient updates, as shown in the Bellman equation in Eq. (1):

$$\mathcal{M}(f_z)(\cdot) = \sum_{z',r} \alpha_{z,z',r} \mathcal{M}(f_{z'}) \left( \frac{\cdot - r}{\gamma} \right).$$

However, for current well-established DGMs, their modeling operators are either not explicitly defined (e.g., VAE and GAN), lacking guaranteed linearity, or are nonlinear operators (e.g., $\mathcal{M}_{\text{EBM}}$, $\mathcal{M}_{\text{Flow}}$, and $\mathcal{M}_{\text{SGM}}$). Consequently, this restricts the application of these powerful DGMs to MDPs. We provide an extended discussion of related work in Appendix A.

## 3 Method: Bellman Diffusion

In this section, we mainly provide an overview of Bellman Diffusion, presenting the usage, with its theoretical details later in Sec. 4. We defer all proofs to Appendix 3.

### 3.1 Scalar and Vector Field Matching

**Field matching.** Suppose we have a finite set of $D$-dimensional samples, with each data $\mathbf{x}$ drawn from the distribution $p_{\text{target}}$. As a generative model, Bellman Diffusion aims to learn both the gradient field $\nabla p_{\text{target}}$ and the scalar field $p_{\text{target}}$. Similar to Fisher divergence (Antolín et al., 2009) for the score function $\nabla \log p_{\text{target}}$, we introduce two divergences for $\nabla p_{\text{target}}$ and $p_{\text{target}}$.

**Definition 3.1** (Field Divergences). Let $p(\cdot)$ and $q(\cdot)$ be continuous probability densities. The discrepancy between the two can be defined as

$$\mathcal{D}_{\text{grad}}\big(p(\cdot), q(\cdot)\big) = \int p(\mathbf{x}) \left\| \nabla p(\mathbf{x}) - \nabla q(\mathbf{x}) \right\|^2 \mathrm{d}\mathbf{x} \tag{3}$$

using the gradient operator $\nabla$ in terms of $\mathbf{x}$, or as

$$\mathcal{D}_{\text{id}}\big(p(\cdot), q(\cdot)\big) = \int p(\mathbf{x})(p(\mathbf{x}) - q(\mathbf{x}))^2 \mathrm{d}\mathbf{x} \tag{4}$$

using the identity operator $\mathbb{I}$. Here, $\|\cdot\|$ denotes the $\ell_2$ norm.

As shown in Appendix B.2, the two measures above are valid statistical measures. These measures are used to empirically estimate the gradient field $\nabla p_{\text{target}}(\mathbf{x})$ and the scalar field $p_{\text{target}}(\mathbf{x})$ from real data $\mathcal{X}$. Furthermore, our modeling defines a modeling operator given by $\mathcal{M}_{\text{Bellman}} := \begin{bmatrix} \nabla \\ \mathbb{I} \end{bmatrix} \colon p_{\text{target}}(\cdot) \mapsto \begin{bmatrix} \nabla p_{\text{target}}(\cdot) \\ p_{\text{target}}(\cdot) \end{bmatrix}$ which is linear in its input.

Similar to SGMs, we parameterize two neural networks, $\mathbf{g}_{\phi}(\mathbf{x})$ and $s_{\varphi}(\mathbf{x}) \geqslant 0$, with learnable parameters $\phi$ and $\varphi$, to match with these fields using the following estimation loss functions:

$$\begin{cases} \mathcal{L}_{\text{grad}}(\phi) := \mathcal{D}_{\text{grad}}\big(p_{\text{target}}(\cdot), \mathbf{g}_{\phi}(\cdot)\big) = \mathbb{E}_{\mathbf{x} \sim p_{\text{target}}(\mathbf{x})} \left[ \left\| \nabla p_{\text{target}}(\mathbf{x}) - \mathbf{g}_{\phi}(\mathbf{x}) \right\|^2 \right] \\ \mathcal{L}_{\text{id}}(\varphi) := \mathcal{D}_{\text{id}}\big(p_{\text{target}}(\cdot), s_{\varphi}(\cdot)\big) = \mathbb{E}_{\mathbf{x} \sim p_{\text{target}}(\mathbf{x})} \left[ (p_{\text{target}}(\mathbf{x}) - s_{\varphi}(\mathbf{x}))^2 \right]. \end{cases} \tag{5}$$

Since the terms $\nabla p_{\text{target}}(\mathbf{x})$ and $p_{\text{target}}(\mathbf{x})$ inside the expectation are generally inaccessible, these losses cannot be estimated via Monte Carlo sampling. The following proposition resolves this issue by deriving a feasible proxy for the loss functions.

**Proposition 3.1** (Equivalent Forms of Field Matching). *The loss $\mathcal{L}_{\mathrm{grad}}(\boldsymbol{\phi})$ is given by*

$$\mathcal{L}_{\mathrm{grad}}(\boldsymbol{\phi}) = C_{\mathrm{grad}} + \lim_{\epsilon \to 0} \mathbb{E}_{\mathbf{x}_1, \mathbf{x}_2 \sim p_{\mathrm{target}}(\mathbf{x})}\Big[\|\mathbf{g}_{\boldsymbol{\phi}}(\mathbf{x}_1)\|^2 + \mathrm{tr}(\nabla \mathbf{g}_{\boldsymbol{\phi}}(\mathbf{x}_1))\mathcal{N}(\mathbf{x}_2 - \mathbf{x}_1; \mathbf{0}, \epsilon \mathbf{I}_D)\Big],$$

*and $\mathcal{L}_{\mathrm{id}}(\boldsymbol{\varphi})$ is expressed as*

$$\mathcal{L}_{\mathrm{id}}(\boldsymbol{\varphi}) = C_{\mathrm{id}} + \lim_{\epsilon \to 0} \mathbb{E}_{\mathbf{x}_1, \mathbf{x}_2 \sim p_{\mathrm{target}}(\mathbf{x})}\Big[s_{\boldsymbol{\varphi}}(\mathbf{x}_1)^2 - 2s_{\boldsymbol{\varphi}}(\mathbf{x}_1)\mathcal{N}(\mathbf{x}_2 - \mathbf{x}_1; \mathbf{0}, \epsilon \mathbf{I}_D)\Big].$$

*Here, $\mathcal{N}(\cdot; \mathbf{0}, \epsilon \mathbf{I}_D)$ denotes a $D$-dimensional isotropic Gaussian density function, and $C_{\mathrm{grad}}$ and $C_{\mathrm{id}}$ are constants independent of the model parameters $\boldsymbol{\phi}$ and $\boldsymbol{\varphi}$.*

We note that using the sequence of isotropic Gaussian densities $\{\mathcal{N}(\cdot; \mathbf{0}, \epsilon \mathbf{I}_D)\}_{\epsilon > 0}$ is not strictly necessary. It is a convenient choice for constructing a family of distributions with parameters $\epsilon > 0$ that approximates the delta distribution as $\epsilon \to 0^+$, enabling feasible and simple training objectives.

Building on the above proposition, we can obtain feasible approximations of the training losses. With $\epsilon$ fixed to be sufficiently small (see Sec. 6 for experimental setups), we have:

$$\begin{cases} \bar{\mathcal{L}}_{\mathrm{grad}}(\boldsymbol{\phi}; \epsilon) := C_{\mathrm{grad}} + \mathbb{E}_{\mathbf{x}_1, \mathbf{x}_2 \sim p_{\mathrm{target}}(\mathbf{x})}\Big[\|\mathbf{g}_{\boldsymbol{\phi}}(\mathbf{x}_1)\|^2 + \mathrm{tr}(\nabla \mathbf{g}_{\boldsymbol{\phi}}(\mathbf{x}_1))\mathcal{N}(\cdot, \mathbf{0}, \epsilon \mathbf{I}_D)\Big] \approx \mathcal{L}_{\mathrm{grad}}(\boldsymbol{\phi}), \\ \bar{\mathcal{L}}_{\mathrm{id}}(\boldsymbol{\varphi}; \epsilon) := C_{\mathrm{id}} + \mathbb{E}_{\mathbf{x}_1, \mathbf{x}_2 \sim p_{\mathrm{target}}(\mathbf{x})}\Big[s_{\boldsymbol{\varphi}}(\mathbf{x}_1)^2 - 2s_{\boldsymbol{\varphi}}(\mathbf{x}_1)\mathcal{N}(\mathbf{x}_2 - \mathbf{x}_1, \mathbf{0}, \epsilon \mathbf{I}_D)\Big] \approx \mathcal{L}_{\mathrm{id}}(\boldsymbol{\varphi}) \end{cases}$$

$$(6)$$

We note that scalar and gradient fields can be modeled independently. Moreover, as Bellman Diffusion directly matches these fields, it eliminates the need for the normalizing constant associated with costly spatial integrals in the density network required by EBMs.

## 3.2 Efficient Field Matching Losses

**Slice trick for efficient training.** While the loss functions $\bar{\mathcal{L}}_{\mathrm{grad}}(\boldsymbol{\phi}; \epsilon)$ and $\bar{\mathcal{L}}_{\mathrm{id}}(\boldsymbol{\varphi}; \epsilon)$ support Monte Carlo estimation, the term $\mathrm{tr}(\nabla \mathbf{g}_{\boldsymbol{\phi}}(\mathbf{x}_1))$ in $\bar{\mathcal{L}}_{\mathrm{grad}}(\boldsymbol{\phi}; \epsilon)$ is computationally expensive, limiting the scalability in high dimensions. To address this problem, we apply the slice trick (Kolouri et al., 2019; Song et al., 2020) to estimate the trace term efficiently. The resulting objective is summarized in the following proposition.

**Proposition 3.2** (Sliced Gradient Matching). *We define the sliced version of $\mathcal{L}_{\mathrm{grad}}$ (i.e., Eq. (3)) as*

$$\mathcal{L}_{\mathrm{grad}}^{\mathrm{slice}}(\boldsymbol{\phi}) = \mathbb{E}_{\mathbf{v} \sim q(\mathbf{v}), \mathbf{x} \sim p_{\mathrm{target}}(\mathbf{x})}\Big[\big(\mathbf{v}^\top \nabla p_{\mathrm{target}}(\mathbf{x}) - \mathbf{v}^\top \mathbf{g}_{\boldsymbol{\phi}}(\mathbf{x})\big)^2\Big],$$

*where $\mathbf{v}$ represents the slice vector drawn from a continuous distribution $q(\mathbf{v})$. This sliced loss also has an equivalent form:*

$$\mathcal{L}_{\mathrm{grad}}^{\mathrm{slice}}(\boldsymbol{\phi}) = C'_{\mathrm{grad}} + \lim_{\epsilon \to 0} \mathbb{E}_{\substack{\mathbf{v} \sim q(\mathbf{v}); \\ \mathbf{x}_1, \mathbf{x}_2 \sim p_{\mathrm{target}}(\mathbf{x})}}\Big[(\mathbf{v}^\top \mathbf{g}_{\boldsymbol{\phi}}(\mathbf{x}_1))^2 + (\mathbf{v}^\top \nabla_{\mathbf{x}_1} \mathbf{g}_{\boldsymbol{\phi}}(\mathbf{x}_1)\mathbf{v})\mathcal{N}(\mathbf{x}_2 - \mathbf{x}_1; \mathbf{0}, \epsilon \mathbf{I}_D)\Big],$$

*where $C'_{\mathrm{grad}}$ is another constant independent of the model parameters.*

Similar to Eq. (6), we can define a proxy loss for $\mathcal{L}_{\mathrm{grad}}^{\mathrm{slice}}(\boldsymbol{\phi})$ as follows with a sufficiently small $\epsilon$:

$$\mathbb{E}_{\mathbf{v} \sim q(\mathbf{v}); \mathbf{x}_1, \mathbf{x}_2 \sim p_{\mathrm{target}}(\mathbf{x})}\Big[(\mathbf{v}^\top \mathbf{g}_{\boldsymbol{\phi}}(\mathbf{x}_1))^2 + (\mathbf{v}^\top \nabla_{\mathbf{x}_1} \mathbf{g}_{\boldsymbol{\phi}}(\mathbf{x}_1)\mathbf{v})\mathcal{N}(\mathbf{x}_2 - \mathbf{x}_1; \mathbf{0}, \epsilon \mathbf{I}_D)\Big], \qquad (7)$$

which allows Monte Carlo estimation from samples $\mathcal{X}$. This proxy loss serves as a reasonable estimator (Lai et al., 2023) for $\mathcal{L}_{\mathrm{grad}}(\boldsymbol{\phi})$.

**Slice trick for improving sample efficiency.** When the data dimension $D$ is large, the multiplier $\mathcal{N}(\mathbf{x}_2 - \mathbf{x}_1; \mathbf{0}, \epsilon \mathbf{I}_D)$ in the loss functions: $\bar{\mathcal{L}}_{\mathrm{grad}}(\boldsymbol{\phi}; \epsilon)$ and $\bar{\mathcal{L}}_{\mathrm{id}}(\boldsymbol{\varphi}; \epsilon)$, will become nearly zero due to the $(2\pi)^{-D/2}$ factor, requiring a very large batch size for accurate Monte Carlo estimation and leading to low data efficiency.

To resolve this issue, we apply an additional slice trick, projecting the $D$-dimensional Gaussian density $\mathcal{N}(\mathbf{x}_2 - \mathbf{x}_1; \mathbf{0}, \epsilon \mathbf{I}_D)$ into a 1-dimensional density $\mathcal{N}(\mathbf{w}^\top \mathbf{x}_2 - \mathbf{w}^\top \mathbf{x}_1, 0, \epsilon)$ along a random direction $\mathbf{w} \sim q(\mathbf{w})$, where $\mathbf{w}$ follows a slice vector distribution $q(\mathbf{w})$. Combining with Eq. (7), this results in our ultimate gradient field matching loss:

$$\bar{\mathcal{L}}_{\text{grad}}^{\text{slice}}(\boldsymbol{\phi}; \epsilon) := \mathbb{E}_{\substack{\mathbf{w} \sim q(\mathbf{w}), \mathbf{v} \sim q(\mathbf{v}); \\ \mathbf{x}_1, \mathbf{x}_2 \sim p_{\text{target}}(\mathbf{x})}} \left[ (\mathbf{v}^\top \mathbf{g}_{\boldsymbol{\phi}}(\mathbf{x}_1))^2 + (\mathbf{v}^\top \nabla_{\mathbf{x}_1} \mathbf{g}_{\boldsymbol{\phi}}(\mathbf{x}_1) \mathbf{v}) \mathcal{N}(\mathbf{w}^\top \mathbf{x}_2 - \mathbf{w}^\top \mathbf{x}_1; 0, \epsilon) \right].$$
$$(8)$$

Similarly, we apply the same trick to $\bar{\mathcal{L}}_{\text{id}}(\boldsymbol{\varphi}; \epsilon)$ for dimension projection and obtain:

$$\bar{\mathcal{L}}_{\text{id}}^{\text{slice}}(\boldsymbol{\varphi}; \epsilon) := \mathbb{E}_{\substack{\mathbf{w} \sim q(\mathbf{w}); \\ \mathbf{x}_1, \mathbf{x}_2 \sim p_{\text{target}}(\mathbf{x})}} \left[ s_{\boldsymbol{\varphi}}(\mathbf{x}_1)^2 - 2 s_{\boldsymbol{\varphi}}(\mathbf{x}_1) \mathcal{N}(\mathbf{w}^\top \mathbf{x}_2 - \mathbf{w}^\top \mathbf{x}_1; 0, \epsilon) \right]. \quad (9)$$

We adopt $\bar{\mathcal{L}}_{\text{grad}}^{\text{slice}}(\boldsymbol{\phi}; \epsilon)$ and $\bar{\mathcal{L}}_{\text{id}}^{\text{slice}}(\boldsymbol{\varphi}; \epsilon)$ for vector and scalar field matching losses, as they offer more practical and efficient objectives than $\mathcal{L}_{\text{grad}}(\boldsymbol{\phi})$ and $\mathcal{L}_{\text{id}}(\boldsymbol{\varphi})$, respectively. Empirically, these adaptations significantly stabilize the model in experiments.

### 3.3 BELLMAN DIFFUSION DYNAMICS

Suppose that neural networks $\mathbf{g}_{\boldsymbol{\phi}}(\mathbf{x}), s_{\boldsymbol{\varphi}}(\mathbf{x})$ accurately estimate the target fields $\nabla p_{\text{target}}(\mathbf{x})$ and $p_{\text{target}}(\mathbf{x})$, one can sample from $p_{\text{target}}(\mathbf{x})$ by approximating the score function as: $\nabla \log p_{\text{target}}(\mathbf{x}) = \frac{\nabla p_{\text{target}}(\mathbf{x})}{p_{\text{target}}(\mathbf{x})} \approx \frac{\mathbf{g}_{\boldsymbol{\phi}}(\mathbf{x})}{s_{\boldsymbol{\varphi}}(\mathbf{x})}$, and then applying Langevin dynamics (Bussi & Parrinello, 2007): $d\mathbf{x}(t) = \nabla \log p_{\text{target}}(\mathbf{x}) dt + \sqrt{2} d\boldsymbol{\omega}(t) \approx \frac{\mathbf{g}_{\boldsymbol{\phi}}(\mathbf{x})}{s_{\boldsymbol{\varphi}}(\mathbf{x})} dt + \sqrt{2} d\boldsymbol{\omega}(t)$, where $\boldsymbol{\omega}(t)$ is a standard Brownian motion. However, this approach can be numerically unstable due to the division[2]. This issue is unavoidable as $p_{\text{target}}(\mathbf{x})$ vanishes when $\|\mathbf{x}\| \to \infty$. Additionally, it doesn't support the distributional Bellman update for MDPs as mentioned in Sec. 1.

To solve this, we propose a new SDE to sample from $p_{\text{target}}$, termed *Bellman Diffusion Dynamics*:

$$d\mathbf{x}(t) = \nabla p_{\text{target}}(\mathbf{x}(t)) dt + \sqrt{p_{\text{target}}(\mathbf{x}(t))} d\boldsymbol{\omega}(t). \quad (10)$$

We also provide the theoretical motivation and derivation of Eq. (10) in Appendix B.1.

In practice, once the neural network approximations $\mathbf{g}_{\boldsymbol{\phi}}(\mathbf{x}) \approx \nabla p_{\text{target}}(\mathbf{x})$ and $s_{\boldsymbol{\varphi}}(\mathbf{x}) \approx p_{\text{target}}(\mathbf{x})$ are both well-learned, we can derive the following *empirical Bellman Diffusion Dynamics*, a feasible proxy SDE for Eq. (10):

$$d\mathbf{x}(t) = \mathbf{g}_{\boldsymbol{\phi}}(\mathbf{x}) dt + \sqrt{s_{\boldsymbol{\varphi}}(\mathbf{x})} d\boldsymbol{\omega}(t). \quad (11)$$

Bellman Diffusion learns and samples using both the scalar and gradient fields, allowing it to better approximate low-density regions and unbalanced target weights (see Sec. 6.1).

### 3.4 SUMMARY OF TRAINING AND SAMPLING ALGORITHMS

To summarize Bellman Diffusion as a DGM, we outline the training and sampling steps in Alg. 3 and Alg. 4 in Appendix D.1. For training, we first sample real data $\mathbf{x}_1, \mathbf{x}_2$ from dataset $\mathcal{X}$ (line 2) and slice vectors $\mathbf{v}, \mathbf{w}$ from some predefined distributions $q(\mathbf{v}), q(\mathbf{w})$ (line 3)[3]. Then, we estimate the loss functions $\bar{\mathcal{L}}_{\text{grad}}^{\text{slice}}(\boldsymbol{\phi}; \epsilon), \bar{\mathcal{L}}_{\text{id}}^{\text{slice}}(\boldsymbol{\varphi}; \epsilon)$ using Monte Carlo sampling (lines 4-6). Finally, the model parameters $\boldsymbol{\phi}$ and $\boldsymbol{\varphi}$ are updated via gradient descent (lines 7-8).

For inference, we begin by sampling $\mathbf{x}(0)$ from an arbitrary distribution, such as standard normal (line 1). Then, after setting the number of steps $T$ and step size $\eta$, we iteratively update $\mathbf{x}(0)$ to $\mathbf{x}(\eta T)$ following Eq. (10) (lines 3-7).

---

[2]For example, if $s_{\boldsymbol{\varphi}}(\mathbf{x})$ is around 0.01, its inverse can magnify the estimation error of $\mathbf{g}_{\boldsymbol{\phi}}(\mathbf{x})$ by 100 times.

[3]Here, we follow the practice in Song et al. (2020) by using a single slice vector to approximate the expectation over $q(\mathbf{v})$ or $q(\mathbf{w})$, trading variance for reduced computational cost.

## 4 MAIN THEORY

In this section, we present theoretical foundations for Bellman Diffusion Dynamics, including steady-state analysis of Eq. (10) and an error analysis for Eq. (11), to justify its underlying rationale. We defer all proofs to Appendix C.

### 4.1 STEADY-STATE ANALYSIS OF BELLMAN DIFFUSION DYNAMICS

Let $p_t$ be the marginal density of Bellman Diffusion Dynamics given by Eq. (10), starting from any initial density $p_0$. The following theorem shows that, regardless of the initial distribution $p_0$, $p_t$ converges to the stationary distribution, which is exactly $p_{\text{target}}$, as $t \to \infty$, at an exponential rate.

**Theorem 4.1** (Convergence to the Steady State). *Let $p_{\text{target}}$ be the target density satisfying Assumption C.1. Then, for any initial density $p_0$, we have the following KL and Wasserstein-2 bounds:*

$$W_2^2\big(p_t, p_{\text{target}}\big) \lesssim \text{KL}\big(p_t \| p_{\text{target}}\big) \lesssim e^{-2\alpha t} \text{KL}\big(p_0 \| p_{\text{target}}\big).$$

*Here, $\alpha > 0$ is some constant determined by $p_{\text{target}}$, and $\lesssim$ hides multiplicative constants that depend only on $p_{\text{target}}$.*

This theorem implies that as $t \to \infty$, $p_t \to p_{\text{target}}$ in both KL and Wasserstein-2 senses. Thus, it justifies that by using our sampling method, which involves solving the SDE in Eq. (10), we can ensure that samples will be obtained from the target distribution $p_{\text{target}}$.

### 4.2 ERROR ANALYSIS OF EMPIRICAL BELLMAN DIFFUSION DYNAMICS

We let $p_{t;\phi,\varphi}$ denote the marginal density from the empirical Bellman Diffusion Dynamics in Eq. (11), starting from any initial density $p_0$. The following theorem extends the result in Theorem 4.1 by providing an error analysis. It accounts for network approximation errors in $\mathbf{g}_\phi(\mathbf{x}) \approx \nabla p_{\text{target}}(\mathbf{x})$ and $s_\varphi(\mathbf{x}) \approx p_{\text{target}}(\mathbf{x})$, and gives an upper bound on the Wasserstein-2 discrepancy between $p_{T;\phi,\varphi}$ and $p_{\text{target}}$.

**Theorem 4.2** (Error Analysis of Neural Network Approximations). *Let $p_{\text{target}}$ be the target distribution satisfying Assumptions C.1 and C.2. Suppose the dynamics in Eqs. (10) and (11) start from the same initial condition sampled from $p_0$. For any $\varepsilon > 0$, if $T = \mathcal{O}(\log 1/\varepsilon^2)$ and $\varepsilon_{\text{est}} = \mathcal{O}\Big(\frac{\varepsilon}{\sqrt{T}e^{\frac{1}{2}LT}}\Big)$, such that $\|g_\phi(\cdot) - \nabla p_{\text{target}}(\cdot)\|_\infty \leqslant \varepsilon_{\text{est}}$ and $|s_\varphi(\cdot) - p_{\text{target}}(\cdot)|_\infty \leqslant \varepsilon_{\text{est}}$, where $L > 0$ is the Lipschitz constant associated with $p_{\text{target}}$, then $W_2(p_{T;\phi,\varphi}, p_{\text{target}}) \leqslant \varepsilon$.*

From the above theorem, our dynamics can function as a standalone generative model, capable of learning the target distribution $p_{\text{target}}$. Using advanced techniques such as Chen et al. (2022); De Bortoli (2022); Kim et al. (2023; 2024), a tighter bound between $p_{T;\phi,\varphi}$ and $p_{\text{target}}$ in $W_2$ or other divergences could be achieved. Moreover, discrete-time versions of both Theorems 4.1 and 4.2 can be derived with more advanced analysis. However, we defer this to future work, as the current focus is on establishing the core principles.

## 5 EXPERIMENTS: BELLMAN DIFFUSION IN DISTRIBUTIONAL RL

We detail the training and evaluation of Bellman Diffusion for distributional RL tasks, demonstrating its effectiveness in this classical MDP setting. A method effective in RL can naturally address simpler MDP tasks, such as planning.

### 5.1 BELLMAN DIFFUSION FOR DISTRIBUTIONAL RL MODELING

An MDP is defined by a 5-tuple $(\mathcal{Z}, \mathcal{A}, p_{\text{tran}}, p_{\text{rwd}}, \gamma)$, where $\mathcal{Z}$ is the state space, $\mathcal{A}$ is the action space, $p_{\text{tran}}$ represents the transition probability, $p_{\text{rwd}}$ is the reward model, and $\gamma$ is the discount factor. Given a policy $\pi$ that selects an action $a \in \mathcal{A}$ for each state $z \in \mathcal{Z}$, the goal is to estimate the probability distribution of the discounted return $X = \sum_{t \geqslant 1} \gamma^{t-1} R_t$ for each state $z$ or state-action pair $(z, a)$, where $R_t$ is the reward received at time step $t$.

**Training and evaluation algorithms.** Let us consider the case of state-action return $X_{z,a}, z \in \mathcal{Z}, a \in \mathcal{A}$. To apply Bellman Diffusion to model this return distribution, we need to first parameterize

| **Algorithm 1** Training with Bellman Diffusion | **Algorithm 2** Inference with Bellman Diffusion |
|---|---|
| 1: **repeat** | 1: Set the initial environment state $z_0 \in \mathcal{Z}$ |
| 2:  Sample state transition $(z_t, a_t, z_{t+1}, a_{t+1}, r_t)$ from the environment and policy $\pi$ | 2: Set the cumulative return $X = 0$, with discount rate $\gamma \in (0, 1)$ |
| 3:  **if** $z_t$ is the end state **then** | 3: Set the current time step $t = 0$ |
| 4:   Sample $x_1, x_2$ from $\mathcal{N}(r_t; 0, \xi)$ | 4: **while** $z_t$ is not the terminal state **do** |
| 5:   Compute $\mathcal{L}_{\mathrm{grad}}(\boldsymbol{\phi}; \epsilon) = g_{\boldsymbol{\phi}}(x_1, z_t, a_t)^2 + \mathcal{N}(x_1 - x_2; 0, \epsilon)\partial_{x_1} g_{\boldsymbol{\phi}}(x_1, z_t, a_t)$ | 5:  Set an empty map $f : \mathcal{A} \to \mathbb{R}$ |
| 6:   Compute $\mathcal{L}_{\mathrm{id}}(\boldsymbol{\varphi}; \epsilon) = s_{\boldsymbol{\varphi}}(x_1, z_t, a_t)^2 - 2\mathcal{N}(x_1 - x_2; 0, \epsilon)s_{\boldsymbol{\varphi}}(x_1, z_t, a_t)$ | 6: **for** $a \in \mathcal{A}$ **do** |
| 7:   Update parameter $\boldsymbol{\phi}$ with $-\nabla_{\boldsymbol{\phi}}\mathcal{L}_{\mathrm{grad}}(\boldsymbol{\phi}; \epsilon)$ | 7:   Sample a batch of particle $x_i$ from uniform distribution $\mathcal{U}(x_{\min}, x_{\max})$ |
| 8:   Update parameter $\boldsymbol{\varphi}$ with $-\nabla_{\boldsymbol{\varphi}}\mathcal{L}_{\mathrm{id}}(\boldsymbol{\varphi}; \epsilon)$ | 8:   Apply the Bellman Diffusion Dynamics (i.e., Eq. (10)), with gradient and scalar fields $g_{\boldsymbol{\phi}}, s_{\boldsymbol{\varphi}}$, to convert each particle $x_i$ into a new one $\bar{x}_i$ |
| 9:  **else** | 9:   Set $f(a)$ as the mean of all new particle $\bar{x}_i$ |
| 10:   Sample $x$ from a bounded span $(x_{\min}, x_{\max})$ | 10: **end for** |
| 11:   Set target gradient $g_{\mathrm{tgt}} = g_{\boldsymbol{\phi}}\left(\frac{x-r}{\gamma}, z_{t+1}, a_{t+1}\right)$ | 11:  Set $a_t = \arg\max_{a \in \mathcal{A}} f(a)$ |
| 12:   Set target scalar $s_{\mathrm{tgt}} = \frac{1}{\gamma} s_{\boldsymbol{\varphi}}\left(\frac{x-r}{\gamma}, z_{t+1}, a_{t+1}\right)$ | 12:  Based on the last state $z_t$ and action $a_t$, get new state $z_{t+1}$ and reward $x$ from the environment |
| 13:   Update $\boldsymbol{\phi}$ with $-\nabla_{\boldsymbol{\phi}}\left(g_{\boldsymbol{\phi}}(x, z_t, a_t) - g_{\mathrm{tgt}}\right)^2$ | 13:  Update the current return $X = x + \gamma X$ |
| 14:   Update $\boldsymbol{\varphi}$ with $-\nabla_{\boldsymbol{\varphi}}\left(s_{\boldsymbol{\varphi}}(x, z_t, a_t) - s_{\mathrm{tgt}}\right)^2$ | 14:  Update time step $t = t + 1$ |
| 15:  **end if** | 15: **end while** |
| 16: **until** parameters $\boldsymbol{\phi}, \boldsymbol{\varphi}$ converge | |

the gradient and scalar fields for every state-action pair $(z, a)$, which is memory consuming. One way to address this inefficiency is to share field models across all state-action pairs. In this spirit, we respectively denote the 1-dimensional gradient and scalar models as $g_{\boldsymbol{\phi}}(x, z, a)$ and $s_{\boldsymbol{\varphi}}(x, z, a)$. Alg. 1 shows the training procedure of field models in terms of the distributional Bellman update, while Alg. 2 shows how to form a policy $\pi : \mathcal{Z} \to \mathcal{A}$ with the field models and evaluate its performance. Alg. 1, together with the 5th-11th lines in Alg. 2 to predict the next action $a_{t+1}$, form a complete distributional RL learning algorithm.

**More algorithm details.** In Alg. 1, we assume that the reward at the terminal state is a scalar and that the variance of the Gaussian $\xi$ is small—an assumption that holds in most scenarios. For instance, at the end of a game, one either wins or loses. We also define $x_{\min}$ and $x_{\max}$ as the minimum and maximum possible returns. This algorithm can also be naturally applied to planning, and as mentioned above, it can also be extended to RL by incorporating action selection: typically choosing the action with the highest expected return, while occasionally exploring randomly. We compare our method to the baseline histogram-based approach C51, which models the return distribution as a simple categorical distribution. Its training algorithm is detailed in Algorithm 1 of their paper.

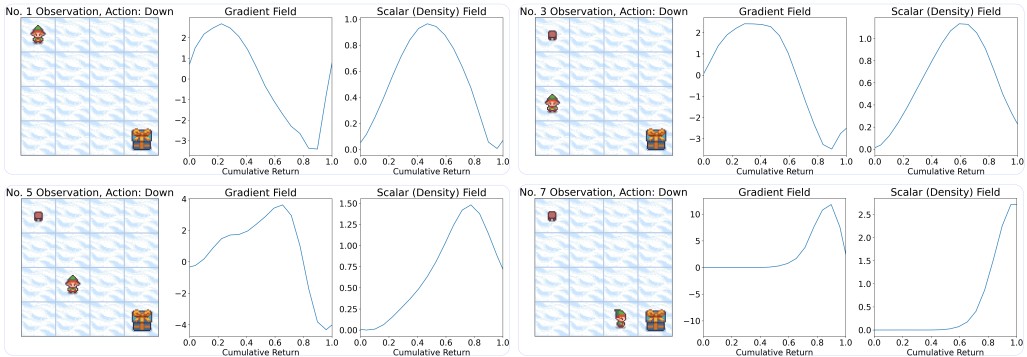

Figure 1: The $2 \times 2$ subfigures, arranged from left to right and top to bottom, show a trajectory of Bellman Diffusion, interacting with a maze environment. Each subfigure consists of the state on the left, gradient field in the middle, and scalar field on the right.

## 5.2 Experimental Results on Distributional RL

We apply Bellman Diffusion to two OpenAI Gym environments (Brockman, 2016): Frozen Lake and Cart Pole. Concrete implementations are detailed in Appendix D.

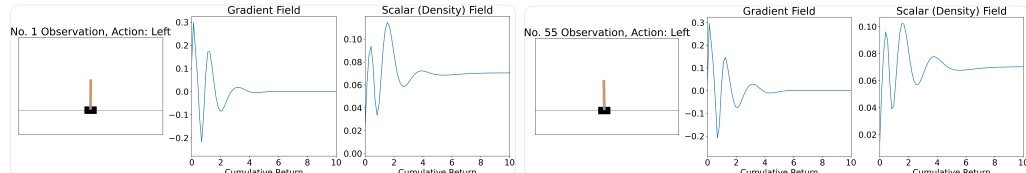

Figure 2: The left and right subfigures respectively show the initial and some middle states of Bellman Diffusion, interacting with an environment of balance control. Every subfigure is composed of the observation on the left, gradient field in the middle, and scalar field on the right.

Frozen Lake is a maze where actions (e.g., moving up) may yield unexpected outcomes (e.g., moving left), while Cart Pole involves balancing a pole on a movable car. Results in Figs. 1 and 2 show that Bellman Diffusion accurately estimates state-level return distributions and their derivatives. For instance, as the agent approaches the goal in the maze, the expected return shifts from $0.5$ to $1$, reflecting that the agent receives no rewards until it reaches the goal.

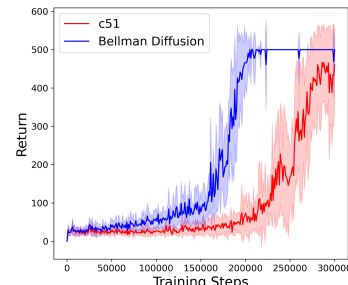

Figure 3: The returns on the Cart Pole, with the colored areas as the confidence interval and the maximum return as $500$.

With the same model sizes and different random seeds, Bellman Diffusion and C51, are both run on the environment of Cart Pole 10 times. The results of return dynamics over training steps are shown in Fig. 3. Both models can ultimately achieve the maximum return; however, Bellman Diffusion converges significantly faster than C51 and exhibits highly stable dynamics. Unlike C51, which accumulates discretization errors across state transitions, our method learns a continuous return distribution, minimizing such errors and achieving superior convergence.

## 6 EXPERIMENTS: BELLMAN DIFFUSION AS A GENERAL DGM

To further verify the effectiveness of Bellman Diffusion as a capable DGM, we conducted extensive experiments on various synthetic and real benchmarks across different tasks. We also place the experiment setup in Appendix D.3 and other supplementary experiments in Appendix E.

### 6.1 SYNTHETIC DATASETS

In this part, we aim to show that Bellman Diffusion can accurately estimate the scalar and gradient fields $\nabla p_{\text{target}}(\mathbf{x}), p_{\text{target}}(\mathbf{x})$ and the associated sampling dynamics can recover the data distribution in terms of the estimation models $\mathbf{g}_\phi(\mathbf{x}), s_\varphi(\mathbf{x})$. For visualization purpose, we will adopt low-dimensional synthetic data (i.e., $D = 1$, or $2$) in the studies.

**1-dimensional uniform distribution.** We use Bellman Diffusion to model a uniform distribution over three disjoint spans, illustrated in the leftmost subfigure of Fig. 4. A key challenge is approximating the discontinuous data distribution using continuous neural networks. Interestingly, the results in Fig. 4 show that the estimated field models $\mathbf{g}_\phi(\mathbf{x})$ and $s_\varphi(\mathbf{x})$ closely match the true values on the support (e.g., $[-1.0, 2.0]$) and perform reasonably in undefined regions. For instance, $\mathbf{g}_\phi(\mathbf{x})$ resembles a negative sine curve on $[-4.5, -0.5]$, aligning with the definitions of one-sided derivative. Notably, Bellman Diffusion Dynamic yields a generated distribution closely aligned with the target distribution and demonstrates its effectiveness in modeling discontinuous data distributions.

**2-Dimensional Mixture of Gaussians (MoG).** Bellman Diffusion effectively approximates the density and gradient fields for multimodal distributions, even with unbalanced weights. We demonstrate this using a MoG with three modes with weights $0.45$, $0.45$, and $0.1$, as shown in the leftmost subfigure of Fig. 5. The right three subfigures show accurate estimations of both scalar and gradient fields for the target distribution and its gradient. The three clustering centers of the training data align with the density peaks in the scalar field (leftmost subfigure) and critical points in the gradient field (middle subfigure). Bellman Diffusion successfully recovers the unbalanced modes of the

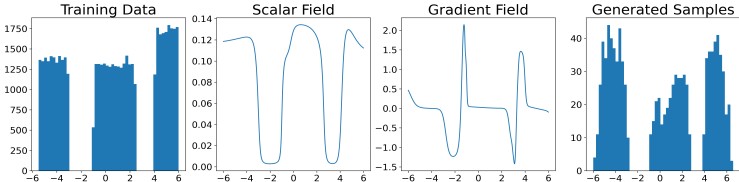

Figure 4: *Bellman Diffusion captures the uniform distribution supported on disjoint spans.* The leftmost subfigure presents the training data histogram, while the next three show the estimated density, derivative functions, and samples generated by Bellman Diffusion.

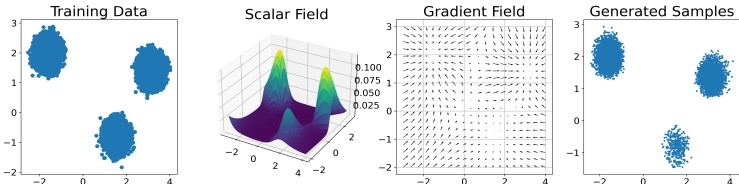

Figure 5: *Bellman Diffusion learns the unbalanced MoG*, which is hard for score-based models. The subfigures, from left to right, display the training data, estimated scalar and gradient fields, and samples generated by our Bellman Diffusion.

target distribution and estimates the fields accurately, even in low-density regions—a challenge for SGMs (Song & Ermon, 2019). Additional results in Appendix E.1 show Bellman Diffusion's effectiveness in generating clustered, geometric data structures using the "moon-shape" dataset, along with a comparison to DDPM (a type of SGM) on MoG datasets.

## 6.2 HIGH-DIMENSIONAL DATA GENERATION

In this section, we follow the common practice (Song et al., 2020) to examine the scalability of our approach across multiple UCI tabular datasets (Asuncion et al., 2007), including Abalone, Telemonitoring, Mushroom, Parkinson's, and Red Wine. We apply several preprocessing steps to these datasets, such as

| Dataset | Denoising Diffusion Models | | Our Model: Bellman Diffusion | |
|---|---|---|---|---|
| | Wasserstein ↓ | MMD ($10^{-3}$) ↓ | Wasserstein ↓ | MMD ($10^{-3}$) ↓ |
| Abalone | 0.975 | 5.72 | 0.763 | 5.15 |
| Telemonitoring | 2.167 | 10.15 | 2.061 | 9.76 |
| Mushroom | 1.732 | 4.29 | 1.871 | 5.12 |
| Parkinsons | 0.862 | 3.51 | 0.995 | 3.46 |
| Red Wine | 1.151 | 3.83 | 1.096 | 3.91 |

Figure 6: *Results on high-dimensional datasets.* Bellman Diffusion is an effective DGM in high dimensions.

imputation and feature selection, resulting in data dimensions of 7, 16, 5, 15, and 10, respectively. For evaluation metrics, we utilize the commonly used Wasserstein distance (Rüschendorf, 1985) and maximum mean discrepancy (MMD) (Dziugaite et al., 2015). The performance of a generative model is considered better when both metrics are lower. Table 6 shows the experimental results. We observe that, regardless of the dataset or metric, Bellman Diffusion performs competitively with DDPM (Ho et al., 2020; Song & Ermon, 2019), a diffusion model known for its scalability.

We further demonstrate in Appendix E.2 that Bellman Diffusion is compatible with VAE (Kingma, 2013), allowing latent generative model training similar to latent diffusion models (Rombach et al., 2022) for higher-resolution image generation. These results demonstrate that Bellman Diffusion is a scalable DGM. However, a more comprehensive study on large-scale Bellman Diffusion as a DGM is left for future work, as our current focus is on unlocking DGM applications in MDPs.

## 7 CONCLUSION

This work addresses the limitations of modern DGMs in MDPs and distributional RL, emphasizing the need for linearity in modelings. We propose Bellman Diffusion, a novel DGM that maintains linearity by modeling gradient and scalar fields. Through new divergence measures and a SDE-based sampling method (Bellman Diffusion Dynamics), we ensure convergence to the target distribution. Experimental results show that Bellman Diffusion provides accurate estimations and outperforms traditional RL methods, offering a promising approach for integrating DGMs into RL frameworks.

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

# Appendix

# A    RELATED WORK

## A.1    RELATED WORK ON DGMS

Deep generative models (DGMs) have gained significant attention in recent years due to their ability to learn complex data distributions and generate high-fidelity samples. This literature review covers several prominent categories of DGMs, including Variational Autoencoders (VAEs), Generative Adversarial Networks (GANs), energy-based methods, flow-based methods, and diffusion models.

**Variational Autoencoders (VAEs).**    Variational Autoencoders (VAEs) are a class of generative models that leverage variational inference to approximate the posterior distribution of latent variables given the data. The VAE framework is based on the evidence lower bound (ELBO), which can be expressed as:

$$\mathcal{L}(\boldsymbol{\theta}, \boldsymbol{\phi}; \mathbf{x}) = \mathbb{E}_{q_{\boldsymbol{\phi}}(\mathbf{z}|\mathbf{x})}[\log p_{\boldsymbol{\theta}}(\mathbf{x}|\mathbf{z})] - \mathcal{D}_{\mathrm{KL}}(q_{\boldsymbol{\phi}}(\mathbf{z}|\mathbf{x})||p(\mathbf{z})),$$

where $q_{\boldsymbol{\phi}}(\mathbf{z}|\mathbf{x})$ is the approximate posterior, $p_{\boldsymbol{\theta}}(\mathbf{x}|\mathbf{z})$ is the likelihood, and $\mathcal{D}_{\mathrm{KL}}$ denotes the Kullback-Leibler divergence. VAEs have shown remarkable success in generating images and other complex data types Kingma (2013).

**Generative Adversarial Networks (GANs).**    Generative Adversarial Networks (GANs) consist of two neural networks, a generator $G$ and a discriminator $D$, that compete against each other. The generator aims to create realistic samples $G(\mathbf{z})$ from random noise $\mathbf{z}$, while the discriminator attempts to distinguish between real samples $\mathbf{x}$ and generated samples $G(\mathbf{z})$. The objective function for GANs can be formulated as:

$$\min_{G} \max_{D} \mathbb{E}_{\mathbf{x} \sim p_{\mathrm{target}}(\mathbf{x})}[\log D(\mathbf{x})] + \mathbb{E}_{\mathbf{z} \sim p_{\mathbf{z}}(\mathbf{z})}[\log(1 - D(G(\mathbf{z})))],$$

where $p_{\mathrm{data}}(\mathbf{x})$ is the data distribution and $p_{\mathbf{z}}(\mathbf{z})$ is the prior distribution on the noise. GANs have become popular for their ability to produce high-quality images and have been applied in various domains Goodfellow et al. (2020).

**Energy-Based Models**    Energy-based models (EBMs) define a probability distribution through an energy function $E(\mathbf{x})$ that assigns lower energy to more probable data points. The probability of a data point is given by:

$$p(\mathbf{x}) = \frac{1}{Z} \exp(-E(\mathbf{x})),$$

where $Z = \int \exp(-E(\mathbf{x}))d\mathbf{x}$ is the partition function. Training EBMs typically involves minimizing the negative log-likelihood of the data (LeCun et al., 2006). They have been successfully applied in generative tasks, including image generation and modeling complex data distributions.

**Flow-Based Methods**    Flow-based methods, such as Normalizing Flows (NFs), learn a bijective mapping between a simple distribution $\mathbf{z}$ and a complex data distribution $\mathbf{x}$ through a series of invertible transformations. The probability density of the data can be expressed as:

$$p(\mathbf{x}) = p(\mathbf{z}) \left| \det \frac{\partial \mathbf{f}^{-1}}{\partial \mathbf{x}} \right|,$$

where $\mathbf{f}$ is the invertible transformation from $\mathbf{z}$ to $\mathbf{x}$. Flow-based models allow for efficient exact likelihood estimation and have shown promise in generating high-quality samples (Rezende & Mohamed, 2015).

**Diffusion Models**    Diffusion models are a class of generative models that learn to generate data by reversing a gradual noising process. The generative process can be described using a stochastic differential equation (SDE):

$$\mathrm{d}\mathbf{x}_t = \mathbf{f}(\mathbf{x}_t, t)\, \mathrm{d}dt + g(t)\, \mathrm{d}d\mathbf{w}_t,$$

where $\mathbf{w}_t$ is a Wiener process, and $\mathbf{f}(\mathbf{x}_t, t)$ and $g(t)$ are functions defining the drift and diffusion terms, respectively. The model learns to recover the data distribution from noise by training on the denoising score matching objective (Ho et al., 2020; Song et al., 2021). Diffusion models have recently gained attention for their impressive image synthesis capabilities.

## A.2 RELATED WORK ON MDPS

Limited by the linearity of the distributional Bellman equation, Previous works (Bellemare et al., 2017; Hessel et al., 2018; Dabney et al., 2018) in planning and distributional RL have relied on conventional generative models to represent state-level return distributions. For instance, the widely used C51 (Bellemare et al., 2017) is a histogram model, resulting in discrete approximation errors. In contrast, Bellman Diffusion is a new type of diffusion model that serves as an expressive distribution approximator without discretization errors.

Recent work (Messaoud et al., 2024) ($S^2$AC) also explores to leverage DGMs for RL tasks. However, they do not align with the problem setting and objectives of our paper. $S^2$AC is designed for Maximum Entropy (MaxEnt) RL, where the primary goal is learning a stochastic policy. In contrast, our work focuses on distributional RL, which aims to model the return distribution for each state. These are fundamentally different RL paradigms, addressing distinct challenges and requiring tailored methodologies. Similar to the Langevin dynamics used in vanilla diffusion models, the SVGD sampler relies on the score function. This dependency introduces challenges when applied to distributional RL, where we aim to model the return distribution without direct reliance on the score of the updated particle.

One concurrent work (Schramm & Boularias, 2024) (BDM) may share some conceptual connections as Bellman Diffusion, they address different problem settings and have distinct goals. BDM is grounded in standard value-based RL and aims to estimate the *successor state measure (SSM)*. In contrast, our work focuses on *distributional RL*, where the goal is to model the entire return distribution for each state, rather than just its first moment (i.e., the value). This difference reflects a fundamental distinction in the types of information each method seeks to capture. Additionally, while BDM estimates the SSM, it does not explicitly discuss how a policy can be derived from it. In comparison, distributional RL, as utilized in our method, provides a direct framework for deriving a feasible policy from the return distribution. This makes our approach more readily applicable to practical RL tasks. At last, BDM focuses primarily on introducing the concept of SSM estimation using diffusion models, but it does not include experimental validation to support its methodology. In contrast, we provide a comprehensive framework with experimental results that demonstrate the effectiveness of Bellman Diffusion in both RL tasks and generative modeling.

# B THEORETICAL RESULTS AND PROOFS FOR SEC. 3

## B.1 MOTIVATION OF THE PROPOSED DYNAMICS IN EQ. (10)

**1-dimensional Case.** Let us first consider the one-dimensional case:

$$\mathrm{d}x(t) = f(x(t))\,\mathrm{d}t + g(x(t))\,\mathrm{d}w(t).$$

Based on the Fokker–Planck equation Risken & Risken (1996), the probability distribution $p(x,t)$ of dynamics $x(t)$ satisfies

$$\frac{\partial p(x,t)}{\partial t} = -\frac{\partial}{\partial x}\Big(f(x)p(x,t)\Big) + \frac{1}{2}\frac{\partial^2}{\partial x^2}\Big(g^2(x)p(x,t)\Big)$$
$$= \frac{\partial}{\partial x}\Big(-f(x)p(x,t) + \frac{1}{2}\frac{\partial}{\partial x}\Big(g^2(x)p(x,t)\Big)\Big).$$

Suppose that the density $p(x,t)$ converges as $t \to \infty$, then we have $\partial p(x,t)/\partial t\,|_{t\to\infty} = 0$. As a result, the above equality indicates that

$$-f(x)p(x,\infty) + \frac{1}{2}\frac{\mathrm{d}}{\mathrm{d}x}\Big(g^2(x)p(x,\infty)\Big) = C.$$

Suppose the constant $C$ is 0, then we have

$$g^2(x)\frac{\mathrm{d}p(x,\infty)}{\mathrm{d}x} = \Big(2f(x) - \frac{\mathrm{d}g^2(x)}{\mathrm{d}x}\Big)p(x,\infty).$$

One way to make this equality hold is to have the below setup:

$$\begin{cases} g^2(x) = p(x,\infty) \\ f(x) = \frac{1}{2}\Big(\frac{\mathrm{d}p(x,\infty)}{\mathrm{d}x} + \frac{\mathrm{d}g^2(x)}{\mathrm{d}x}\Big)\Big) = \frac{\mathrm{d}p(x,\infty)}{\mathrm{d}x}. \end{cases}$$

Therefore, the following dynamics:

$$\mathrm{d}x(t) = \frac{\mathrm{d}p_{\mathrm{target}}(x(t))}{\mathrm{d}x}\,\mathrm{d}t + \sqrt{p_{\mathrm{target}}(x(t))}\,\mathrm{d}w(t),$$

will converge to distribution $p_{\mathrm{target}}(x)$ as $t \to \infty$, regardless of the initial distribution $p(x, 0)$.

$D$**-dimensional Case.**   For the general situation, the dynamics will be

$$\mathrm{d}\mathbf{x}(t) = \nabla_{\mathbf{x}(t)}p_{\mathrm{target}}(\mathbf{x}(t))\,\mathrm{d}t + \sqrt{p_{\mathrm{target}}(\mathbf{x}(t))}\,\mathrm{d}\boldsymbol{\omega}(t),$$

Let us check this expression. Firstly, the Fokker–Planck equation indicates that

$$\frac{\partial p(\mathbf{x}, t)}{\partial t} = -\nabla_{\mathbf{x}} \cdot \big(p(\mathbf{x}, t)\nabla_{\mathbf{x}}p_{\mathrm{target}}(\mathbf{x})\big) + \frac{1}{2}\Delta_{\mathbf{x}}\big(p_{\mathrm{target}}(\mathbf{x})p(\mathbf{x}, t)\big)$$

$$= \nabla_{\mathbf{x}} \cdot \Big(-p(\mathbf{x}, t)\nabla_{\mathbf{x}}p_{\mathrm{target}}(\mathbf{x}) + \frac{1}{2}\nabla_{\mathbf{x}}\big(p_{\mathrm{target}}(\mathbf{x})p(\mathbf{x}, t)\big)\Big),$$

where $\Delta_{\mathbf{x}} = \nabla_{\mathbf{x}} \cdot \nabla_{\mathbf{x}}$ is the Laplace operator. With the Leibniz rule, we have

$$-p(\mathbf{x}, t)\nabla_{\mathbf{x}}p_{\mathrm{target}}(\mathbf{x}) + \frac{1}{2}\nabla_{\mathbf{x}}\big(p_{\mathrm{target}}(\mathbf{x})p(\mathbf{x}, t)\big)$$

$$= -p(\mathbf{x}, t)\nabla_{\mathbf{x}}p_{\mathrm{target}}(\mathbf{x}) + \frac{1}{2}p(\mathbf{x}, t)\nabla_{\mathbf{x}}p_{\mathrm{target}}(\mathbf{x}) + \frac{1}{2}p_{\mathrm{target}}(\mathbf{x})\nabla_{\mathbf{x}}p(\mathbf{x}, t)$$

$$= \frac{1}{2}p_{\mathrm{target}}(\mathbf{x})\nabla_{\mathbf{x}}p(\mathbf{x}, t) - \frac{1}{2}p(\mathbf{x}, t)\nabla_{\mathbf{x}}p_{\mathrm{target}}(\mathbf{x}).$$

Combining the above two equations, we get

$$\frac{\partial p(\mathbf{x}, t)}{\partial t} = \frac{1}{2}\nabla_{\mathbf{x}} \cdot \Big(p_{\mathrm{target}}(\mathbf{x})\nabla_{\mathbf{x}}p(\mathbf{x}, t) - p(\mathbf{x}, t)\nabla_{\mathbf{x}}p_{\mathrm{target}}(\mathbf{x})\Big).$$

By applying the Leibniz rule to divergence operators, we have

$$\begin{cases} \nabla_{\mathbf{x}} \cdot \big(p_{\mathrm{target}}(\mathbf{x})\nabla_{\mathbf{x}}p(\mathbf{x}, t)\big) = \big\langle\nabla_{\mathbf{x}}p_{\mathrm{target}}(\mathbf{x}), \nabla_{\mathbf{x}}p(\mathbf{x}, t)\big\rangle + p_{\mathrm{target}}(\mathbf{x})\Delta_{\mathbf{x}}p(\mathbf{x}, t) \\ \nabla_{\mathbf{x}} \cdot \big(p(\mathbf{x}, t)\nabla_{\mathbf{x}}p_{\mathrm{target}}(\mathbf{x})\big) = \big\langle\nabla_{\mathbf{x}}p(\mathbf{x}, t), \nabla_{\mathbf{x}}p_{\mathrm{target}}(\mathbf{x})\big\rangle + p(\mathbf{x}, t)\Delta_{\mathbf{x}}p_{\mathrm{target}}(\mathbf{x}) \end{cases}.$$

Therefore, the original partial differential equation (PDE) can be simplified as

$$\frac{\partial p(\mathbf{x}, t)}{\partial t} = \frac{1}{2}\Big(p_{\mathrm{target}}(\mathbf{x})\Delta_{\mathbf{x}}p(\mathbf{x}, t) - p(\mathbf{x}, t)\Delta_{\mathbf{x}}p_{\mathrm{target}}(\mathbf{x})\Big).$$

Since the dynamics will converge, we set $p_{\mathrm{target}}(\mathbf{x}) = p(\mathbf{x}, \infty)$. Then, we get

$$\frac{\partial p(\mathbf{x}, t)}{\partial t}\Big|_{t \to \infty} = \frac{1}{2}\Big(p(\mathbf{x}, \infty)\Delta_{\mathbf{x}}p(\mathbf{x}, \infty) - p(\mathbf{x}, \infty)\Delta_{\mathbf{x}}p(\mathbf{x}, \infty)\Big) = 0.$$

Therefore, the dynamics lead to sampling from a given distribution $p_{\mathrm{target}}(\mathbf{x})$.

### B.2   VALIDITY OF FIELD DIVERGENCES.

The first step is to check whether $\mathcal{D}_{\mathrm{grad}}, \mathcal{D}_{\mathrm{id}}$ are well defined divergence measures. To this end, we have the below conclusion.

**Theorem B.1** (Well-defined Divergences)**.** *Suppose that $p(\cdot), q(\cdot)$ are probability densities that are second-order continuously differentiable (i.e., in $\mathcal{C}^2$) and that $p(\mathbf{x}) \neq 0$ for all $\mathbf{x}$. Then the divergence measure $\mathcal{D}_{\mathrm{grad}}(p(\cdot), q(\cdot))$ defined by Eq. (3) and that $\mathcal{D}_{\mathrm{id}}(p(\cdot), q(\cdot))$ formulated by Eq. (4) are both valid statistical divergence measures, satisfying the following three conditions:*

- *Non-negativity: $\mathcal{D}_*(p(\cdot), q(\cdot))$ is either zero or positive;*

- *Null condition: $\mathcal{D}_*(p(\cdot), q(\cdot)) = 0$ if and only if $p(\mathbf{x}) = q(\mathbf{x})$ for every point $\mathbf{x}$;*

- *Positive definiteness: $\mathcal{D}_*(p(\cdot), p(\cdot) + \delta p(\cdot))$ is a positive-definite quadratic form for any infinitesimal displacement $\delta p(\cdot)$ from $p(\cdot)$.*

*Here the subscript $*$ represents either* grad *or* id.

*Proof.* Non-negativity condition obviously holds for both $D_{\text{grad}}$ and $D_{\text{id}}$, due to their definitions.

For the null condition,

$$D_{\text{grad}}(p(\cdot), q(\cdot)) = 0 \quad \text{implies} \quad p(\mathbf{x})\|\nabla p(\mathbf{x}) - \nabla q(\mathbf{x})\|^2 = 0 \text{ for all } \mathbf{x}.$$

This implies $\nabla p(\mathbf{x}) = \nabla q(\mathbf{x})$ for all $x$. Since the gradients are equal, $p(\mathbf{x})$ and $q(\mathbf{x})$ differ by at most a constant. For probability densities, this constant must be zero, so $p(\mathbf{x}) = q(\mathbf{x})$. On the other hand, for $D_{\text{id}}(p(\cdot), q(\cdot)) = 0$:

$$D_{\text{id}}(p(\cdot), q(\cdot)) = 0 \quad \text{implies} \quad (p(\mathbf{x}) - q(\mathbf{x}))^2 = 0 \text{ for all } \mathbf{x}$$

This directly implies that $p(\mathbf{x}) = q(\mathbf{x})$ for all $\mathbf{x}$. Hence, both $D_{\text{grad}}$ and $D_{\text{id}}$ satisfy the null condition: $D_*(p(\cdot), q(\cdot)) = 0$ if and only if $p(\mathbf{x}) = q(\mathbf{x})$ for all $\mathbf{x}$.

At last, we prove that the two measurements satisfy the positive definiteness condition. For $D_{\text{grad}}(p(\cdot), p(\cdot) + \delta p(\cdot))$:

$$D_{\text{grad}}(p(\cdot), p(\cdot) + \delta p(\cdot)) = \int p(\mathbf{x})\|\nabla p(\mathbf{x}) - \nabla(p(\mathbf{x}) + \delta p(\mathbf{x}))\|^2 \, \mathrm{d}\mathbf{x} = \int p(\mathbf{x})\|\nabla \delta p(\mathbf{x})\|^2 \, \mathrm{d}\mathbf{x}.$$

This expression is quadratic in $\delta p(\mathbf{x})$, and since norms are positive definite, $D_{\text{grad}}$ is positive definite for any infinitesimal displacement $\delta p(\mathbf{x})$. On the other hand, for $D_{\text{id}}(p(\cdot), p(\cdot) + \delta p(\cdot))$:

$$D_{\text{id}}(p(\cdot), p(\cdot) + \delta p(\cdot)) = \int p(\mathbf{x})(p(\mathbf{x}) - (p(\mathbf{x}) + \delta p(\mathbf{x})))^2 \, \mathrm{d}\mathbf{x} = \int p(\mathbf{x})(\delta p(\mathbf{x}))^2 \, \mathrm{d}\mathbf{x}.$$

Again, this is quadratic in $\delta p(\mathbf{x})$, making $D_{\text{id}}$ positive definite for any infinitesimal displacement $\delta p(\mathbf{x})$. Thus, both $D_{\text{grad}}$ and $D_{\text{id}}$ are positive-definite quadratic forms for any infinitesimal displacement $\delta p(\mathbf{x})$ from $p(\mathbf{x})$. This concludes the proof. $\qquad\square$

Since measures $\mathcal{D}_{\text{grad}}, \mathcal{D}_{\text{id}}$ are well defined, it is valid to derive the corresponding loss functions $\mathcal{L}_{\text{grad}}, \mathcal{L}_{\text{id}}$ as formulated in Eq. (5).

### B.3 PROOF TO PROPOSITION 3.1.

We aim to rearrange the following loss function:

$$\mathcal{L}_{\text{grad}}(\boldsymbol{\phi}) = \mathcal{D}_{\text{grad}}\big(p_{\text{target}}(\cdot), \mathbf{g}_{\boldsymbol{\phi}}(\cdot)\big) = \mathbb{E}_{\mathbf{x} \sim p_{\text{target}}(\mathbf{x})}\Big[\|\nabla p_{\text{target}}(\mathbf{x}) - \mathbf{g}_{\boldsymbol{\phi}}(\mathbf{x})\|^2\Big]$$

By expanding the inner quadratic form, we get

$$\mathcal{L}_{\text{grad}}(\boldsymbol{\phi}) = \int p_{\text{data}}(\mathbf{x})\|\nabla_{\mathbf{x}} p_{\text{data}}(\mathbf{x})\|^2 d\mathbf{x} + \int p_{\text{data}}(\mathbf{x})\|\mathbf{g}_{\boldsymbol{\phi}}(\mathbf{x})\|^2 d\mathbf{x}$$
$$- 2 \int p_{\text{data}}(\mathbf{x})\big(\mathbf{g}_{\boldsymbol{\phi}}(\mathbf{x})\big)^{\top} \nabla_{\mathbf{x}} p_{\text{data}}(\mathbf{x}) d\mathbf{x}.$$

The first term in the right hand side is in fact a constant and we denote it as $C_{\text{grad}}$. Then, by applying the technique of integral by parts, we can simply the last term as

$$2 \int p_{\text{data}}(\mathbf{x})\big(\mathbf{g}_{\boldsymbol{\phi}}(\mathbf{x})\big)^{\top} \nabla_{\mathbf{x}} p_{\text{data}}(\mathbf{x}) d\mathbf{x}$$
$$= \int \big(\mathbf{g}_{\boldsymbol{\phi}}(\mathbf{x})\big)^{\top} \nabla_{\mathbf{x}} p_{\text{data}}(\mathbf{x})^2 d\mathbf{x}$$
$$= \int \nabla_{\mathbf{x}} \cdot \big(p_{\text{data}}(\mathbf{x})^2 \mathbf{g}_{\boldsymbol{\phi}}(\mathbf{x})\big) d\mathbf{x} - \int p_{\text{data}}(\mathbf{x})^2 \big(\nabla_{\mathbf{x}} \cdot \mathbf{g}_{\boldsymbol{\phi}}(\mathbf{x})\big) d\mathbf{x}$$

Suppose the integral area is $\Omega$ (say by taking it as a ball with a radius $R > 0$) and applying Gauss's Divergence Theorem, we have

$$\int_{\Omega} \nabla_{\mathbf{x}} \cdot \big(p_{\text{data}}(\mathbf{x})^2 \mathbf{g}_{\boldsymbol{\phi}}(\mathbf{x})\big) d\mathbf{x} = \int_{\partial\Omega} \mathbf{n}(\mathbf{x})^{\top} \big(p_{\text{data}}(\mathbf{x})^2 \mathbf{g}_{\boldsymbol{\phi}}(\mathbf{x})\big) d\mathbf{x},$$

where $\partial\Omega$ denotes the boundary of area $\Omega$ and $\mathbf{n}(\mathbf{x})$ represents the unit norm to the boundary $\partial\Omega$. Furthermore, suppose that $\lim_{\|\mathbf{x}\|\to\infty} p_{\text{data}}(\mathbf{x}) \to 0$ and $\mathbf{g}_\phi(\mathbf{x})$ are uniformly bounded in $\mathbf{x}$, then this integral vanishes. So, we have the reduced objective:

$$\mathcal{L}_{\text{grad}}(\phi) = C_{\text{grad}} + \int p_{\text{data}}(\mathbf{x})\|\mathbf{g}_\phi(\mathbf{x})\|^2 d\mathbf{x} + \int p_{\text{data}}(\mathbf{x})^2 \text{tr}(\nabla\mathbf{g}_\phi(\mathbf{x}))d\mathbf{x}.$$

For the second integral, we apply the decoupling trick:

$$\int p_{\text{target}}(\mathbf{x})^2 s_\varphi(\mathbf{x})\,\mathrm{d}\mathbf{x}$$

$$= \int p_{\text{target}}(\mathbf{x})p_{\text{target}}(\mathbf{y})s_\varphi(\mathbf{x})\delta(\mathbf{y} - \mathbf{x})\,\mathrm{d}\mathbf{x}\,\mathrm{d}\mathbf{y}$$

$$= \mathbb{E}_{\mathbf{x}\sim p_{\text{target}}(\mathbf{x}),\mathbf{y}\sim p_{\text{target}}(\mathbf{y})}\Big[s_\varphi(\mathbf{x})\delta(\mathbf{y} - \mathbf{x})\Big]$$

$$= \lim_{\epsilon\to0}\mathbb{E}_{\mathbf{x}_1,\mathbf{x}_2\sim p_{\text{target}}(\mathbf{x})}\Big[s_\varphi(\mathbf{x}_1)\mathcal{N}(\mathbf{x}_2 - \mathbf{x}_1;\mathbf{0},\epsilon\mathbf{I}_D)\Big].$$

Here, we use that $\mathcal{N}(\cdot;\mathbf{0},\epsilon\mathbf{I}_D)$ weakly converges to $\delta(\cdot)$ as $\epsilon \to 0^+$. Therefore, we simplify the loss function as

$$\mathcal{L}_{\text{grad}}(\phi) = C_{\text{grad}} + \lim_{\epsilon\to0}\mathbb{E}_{\mathbf{x}_1,\mathbf{x}_2\sim p_{\text{target}}(\mathbf{x})}\Big[\|\mathbf{g}_\phi(\mathbf{x}_1)\|^2 + \text{tr}(\nabla\mathbf{g}_\phi(\mathbf{x}_1))\mathcal{N}(\mathbf{x}_2 - \mathbf{x}_1;\mathbf{0},\epsilon\mathbf{I}_D)\Big],$$

Next, we prove for the case of

$$\mathcal{L}_{\text{id}}(\varphi) = C_{\text{id}} + \lim_{\epsilon\to0}\mathbb{E}_{\mathbf{x}_1,\mathbf{x}_2\sim p_{\text{target}}(\mathbf{x})}\Big[s_\varphi(\mathbf{x}_1)^2 - 2s_\varphi(\mathbf{x}_1)\mathcal{N}(\mathbf{x}_2 - \mathbf{x}_1;\mathbf{0},\epsilon\mathbf{I}_D)\Big],$$

by following a similar argument. Suppose that we have a divergence loss function:

$$\mathcal{L}_{\text{id}}(\varphi) = \int p_{\text{target}}(\mathbf{x})\Big(p_{\text{target}}(\mathbf{x}) - s_\varphi(\mathbf{x})\Big)^2\,\mathrm{d}\mathbf{x}.$$

Then, we can expand the term as

$$\mathcal{L}_{\text{id}}(\varphi) = \int p_{\text{target}}(\mathbf{x})^3\,\mathrm{d}\mathbf{x} - 2\int p_{\text{target}}(\mathbf{x})^2 s_\varphi(\mathbf{x})\,\mathrm{d}\mathbf{x} + \int p_{\text{target}}(\mathbf{x})s_\varphi(\mathbf{x})^2\,\mathrm{d}\mathbf{x}.$$

For the second integral, we apply the trick again:

$$\int p_{\text{target}}(\mathbf{x})^2 s_\varphi(\mathbf{x})\,\mathrm{d}\mathbf{x} = \int p_{\text{target}}(\mathbf{x})p_{\text{target}}(\mathbf{y})s_\varphi(\mathbf{x})\delta(\mathbf{y} - \mathbf{x})\,\mathrm{d}\mathbf{x}\,\mathrm{d}\mathbf{y}$$

$$= \mathbb{E}_{\mathbf{x}\sim p_{\text{target}}(\mathbf{x}),\mathbf{y}\sim p_{\text{target}}(\mathbf{y})}\Big[s_\varphi(\mathbf{x})\delta(\mathbf{y} - \mathbf{x})\Big]$$

$$= \lim_{\epsilon\to0}\mathbb{E}_{\mathbf{x}_1,\mathbf{x}_2\sim p_{\text{target}}(\mathbf{x})}\Big[s_\varphi(\mathbf{x}_1)\mathcal{N}(\mathbf{x}_2 - \mathbf{x}_1;\mathbf{0},\epsilon\mathbf{I}_D)\Big].$$

Here, we use that $\mathcal{N}(\cdot;0,\epsilon\mathbf{I}_D)$ weakly converges to $\delta(\cdot)$ as $\epsilon \to 0$. Therefore, we simplify the loss function as

$$\mathcal{L}_{\text{id}}(\varphi) = C_{\text{id}} + \mathbb{E}_{\mathbf{x}\sim p_{\text{target}}(\mathbf{x})}\Big[s_\varphi(\mathbf{x})^2\Big] - 2\lim_{\epsilon\to0}\mathbb{E}_{\mathbf{x}_1,\mathbf{x}_2\sim p_{\text{target}}(\mathbf{x})}\Big[s_\varphi(\mathbf{x}_1)\mathcal{N}(\mathbf{x}_2 - \mathbf{x}_1;\mathbf{0},\epsilon\mathbf{I}_D)\Big],$$

where $C_{\text{id}}$ is a constant without learnable parameter $\varphi$.

### B.4 PROOF TO PROPOSITION 3.2

*Proof.* We recall the sliced version of $\mathcal{L}_{\text{grad}}$ as:

$$\mathcal{L}_{\text{grad}}^{\text{slice}}(\phi) = \mathbb{E}_{\mathbf{v}\sim q(\mathbf{v}),\mathbf{x}\sim p_{\text{target}}(\mathbf{x})}\Big[\big(\mathbf{v}^\top\nabla p_{\text{target}}(\mathbf{x}) - \mathbf{v}^\top\mathbf{g}_\phi(\mathbf{x})\big)^2\Big].$$

By expanding the quadratic term $(\cdot)^2$ inside the recursive expectations, we have

$$\mathcal{L}_{\text{grad}}^{\text{slice}} = \mathbb{E}_{\mathbf{v}}\Big[\int p_{\text{target}}(\mathbf{x})(\mathbf{v}^\top\nabla_{\mathbf{x}}\mathbf{g}_\phi(\mathbf{x}))^2\,\mathrm{d}\mathbf{x} - 2\int p_{\text{target}}(\mathbf{x})(\mathbf{v}^\top\nabla_{\mathbf{x}}p_{\text{target}}(\mathbf{x}))(\mathbf{v}^\top\nabla_{\mathbf{x}}\mathbf{g}_\phi(\mathbf{x}))\,\mathrm{d}\mathbf{x} + C_{\text{grad}}'\Big],$$

where, $C'_{\mathrm{grad}} := \int p_{\mathrm{target}}(\mathbf{x})(\mathbf{v}^\top \nabla_{\mathbf{x}} p_{\mathrm{target}}(\mathbf{x}))^2 \, \mathrm{d}\mathbf{x}$ is a constant independent of trainable parameter. We will further simplify the second term came from the cross product as:

$$\int p_{\mathrm{target}}(\mathbf{x})(\mathbf{v}^\top \nabla_{\mathbf{x}} p_{\mathrm{target}}(\mathbf{x}))(\mathbf{v}^\top \nabla_{\mathbf{x}} \mathbf{g}_\phi(\mathbf{x})) \, \mathrm{d}\mathbf{x}$$

$$= \frac{1}{2} \int (\mathbf{v}^\top \nabla_{\mathbf{x}} p_{\mathrm{target}}(\mathbf{x})^2)(\mathbf{v}^\top \nabla_{\mathbf{x}} \mathbf{g}_\phi(\mathbf{x})) \, \mathrm{d}\mathbf{x}$$

$$= \frac{1}{2} \int \left( \nabla_{\mathbf{x}} p_{\mathrm{target}}(\mathbf{x})^2 \right)^\top \left( (\mathbf{v}^\top \nabla_{\mathbf{x}} \mathbf{g}_\phi(\mathbf{x}))\mathbf{v} \right) \, \mathrm{d}\mathbf{x}.$$

Note that we can replace the gradient field $\nabla_{\mathbf{x}} \mathbf{g}_\phi(\mathbf{x})$ with neural network $\mathbf{g}_\phi(\mathbf{x})$. By applying the integration by parts, this equality can be expanded as

$$\frac{1}{2} \int \nabla_{\mathbf{x}} \cdot \left( p_{\mathrm{target}}(\mathbf{x})^2 (\mathbf{v}^\top \mathbf{g}_\phi(\mathbf{x}))\mathbf{v} \right) \mathrm{d}\mathbf{x} - \frac{1}{2} \int p_{\mathrm{target}}(\mathbf{x})^2 \nabla_{\mathbf{x}} \cdot \left( (\mathbf{v}^\top \mathbf{g}_\phi(\mathbf{x}))\mathbf{v} \right) \mathrm{d}\mathbf{x}.$$

Let us first handle the first term in the above equation. Applying Gauss's divergence theorem to a ball $\mathbb{B}(R)$ centered at the origin with radius $R > 0$, we get

$$\int_{\mathbb{B}(R)} \nabla_{\mathbf{x}} \cdot \left( p_{\mathrm{target}}(\mathbf{x})^2 (\mathbf{v}^\top \mathbf{g}_\phi(\mathbf{x}))\mathbf{v} \right) \mathrm{d}\mathbf{x} = \int_{\partial \mathbb{B}(R)} \mathbf{n}(\mathbf{x})^\top \left( p_{\mathrm{target}}(\mathbf{x})^2 (\mathbf{v}^\top \mathbf{g}_\phi(\mathbf{x}))\mathbf{v} \right). \quad (12)$$

where $\mathbf{n}(\mathbf{x})$ is the unit norm vector to the region boundary $\partial \mathbb{B}(R)$. Suppose that $p_{\mathrm{target}}(\mathbf{x})$ decays sufficiently fast as $\|\mathbf{x}\|_2 \to \infty$, for instance, $\lim_{\|\mathbf{x}\|_2 \to \infty} p_{\mathrm{target}}(\mathbf{x})/\|\mathbf{x}\|_2^D = 0$ (see Assumption C.1 (iii)), then this term vanishes as $R \to \infty$.

For the second term in the expansion, we have

$$\nabla_{\mathbf{x}} \cdot \left( (\mathbf{v}^\top \mathbf{g}_\phi(\mathbf{x}))\mathbf{v} \right) = \sum_{1 \leqslant i \leqslant D} \frac{\partial((\mathbf{v}^\top \mathbf{g}_\phi(\mathbf{x}))v_i)}{\partial x_i} = \sum_{1 \leqslant i \leqslant D} \sum_{1 \leqslant j \leqslant D} \frac{v_i v_j \mathbf{g}_{\phi,j}(\mathbf{x})}{\partial x_i} = \mathbf{v}^\top \nabla_{\mathbf{x}} \mathbf{g}_\phi(\mathbf{x}) \mathbf{v}.$$

Here, we write $\mathbf{v} = (v_i)_{1 \leqslant i \leqslant D}$ and $\mathbf{x} = (x_i)_{1 \leqslant i \leqslant D}$. Collecting the above derivations, we have

$$\int p_{\mathrm{target}}(\mathbf{x})(\mathbf{v}^\top \nabla_{\mathbf{x}} p_{\mathrm{target}}(\mathbf{x}))(\mathbf{v}^\top \nabla_{\mathbf{x}} \mathbf{g}_\phi(\mathbf{x})) \, \mathrm{d}\mathbf{x} = -\frac{1}{2} \int p_{\mathrm{target}}(\mathbf{x})^2 (\mathbf{v}^\top \nabla_{\mathbf{x}} \mathbf{g}_\phi(\mathbf{x})\mathbf{v}) \, \mathrm{d}\mathbf{x}. \quad (13)$$

Therefore, the loss function can be converted into

$$\mathcal{L}_{\mathrm{grad}}^{\mathrm{slice}} = \mathbb{E}_{\mathbf{v}} \left[ \int p_{\mathrm{target}}(\mathbf{x})(\mathbf{v}^\top \nabla_{\mathbf{x}} \mathbf{g}_\phi(\mathbf{x}))^2 \, \mathrm{d}\mathbf{x} + \int p_{\mathrm{target}}(\mathbf{x})^2 (\mathbf{v}^\top \nabla_{\mathbf{x}} g_\theta(\mathbf{x})\mathbf{v}) \, \mathrm{d}\mathbf{x} \right] + C'_{\mathrm{grad}}. \quad (14)$$

We apply the same trick from the proof of Proposition 3.1—using Dirac expansion—to enable Monte Carlo estimation for the second inner term:

$$\int p_{\mathrm{target}}(\mathbf{x})^2 \left( \mathbf{v}^\top \nabla_{\mathbf{x}} g_\theta(\mathbf{x})\mathbf{v} \right) \mathrm{d}\mathbf{x}$$

$$= \int p_{\mathrm{target}}(\mathbf{x}) \left( \int p_{\mathrm{target}}(\mathbf{y})\delta(\mathbf{y} - \mathbf{x})d\mathbf{y} \right) \left( \mathbf{v}^\top \nabla_{\mathbf{x}} g_\theta(\mathbf{x})\mathbf{v} \right) \mathrm{d}\mathbf{x}$$

$$= \int p_{\mathrm{target}}(\mathbf{x}_1) p_{\mathrm{target}}(\mathbf{x}_2)(\mathbf{v}^\top \nabla_{\mathbf{x}_1} g_\theta(\mathbf{x}_1)\mathbf{v})\delta(\mathbf{x}_2 - \mathbf{x}_1) \, \mathrm{d}\mathbf{x}_1 \, \mathrm{d}\mathbf{x}_2$$

$$= \mathbb{E}_{\mathbf{x}_1, \mathbf{x}_2 \sim p_{\mathrm{target}}(\mathbf{x})} \left[ (\mathbf{v}^\top \nabla_{\mathbf{x}_1} g_\theta(\mathbf{x}_1)\mathbf{v})\delta(\mathbf{x}_2 - \mathbf{x}_1) \right].$$

Combining the above two identities, we have

$$\mathcal{L}_{\mathrm{grad}}^{\mathrm{slice}} = \mathbb{E}_{\mathbf{v} \sim p_{\mathrm{slice}}(\mathbf{v}), \mathbf{x}_1, \mathbf{x}_2 \sim p_{\mathrm{target}}(\mathbf{x})} \left[ (\mathbf{v}^\top g_\theta(\mathbf{x}_1))^2 + (\mathbf{v}^\top \nabla_{\mathbf{x}_1} g_\theta(\mathbf{x}_1)\mathbf{v})\delta(\mathbf{x}_2 - \mathbf{x}_1) \right] + C'_{\mathrm{grad}}, \quad (15)$$

which completes the proof. $\qquad \square$

## C    PROOFS FOR SEC. 4

### C.1    PREREQUISITES FOR THEORETICAL ANALYSIS.

We introduce some notations and terminologies. We recall the definition of *KL divergence* between $p_{\text{target}}$ and density $p$ as

$$\text{KL}\big(p\|p_{\text{target}}\big) := \int_{\mathbb{R}^D} p(\mathbf{x}) \log \frac{p(\mathbf{x})}{p_{\text{target}}(\mathbf{x})}\, d\mathbf{x}.$$

*Fisher divergence* between $p_{\text{target}}$ and $p$ is defined as:

$$J_{p_{\text{target}}}(p) := \int_{\mathbb{R}^D} p(\mathbf{x}) \left\| \nabla_x \log \frac{p(\mathbf{x})}{p_{\text{target}}(\mathbf{x})} \right\|^2 d\mathbf{x}.$$

*Wasserstein-2 distance* ($W_2$) between $p_{\text{target}}$ and $p$ is defined as:

$$W_2^2(p, p_{\text{target}}) := \inf_{\gamma \sim \Gamma(\mu,\nu)} \mathbb{E}_{(\mathbf{x},\mathbf{y}) \sim \gamma} \|\mathbf{x} - \mathbf{y}\|_2^2,$$

where $\Gamma(\mu, \nu)$ is the set of all couplings of $(\mu, \nu)$.

The following summarizes the two assumptions for our main theorems in Sec. 4.

**Assumption C.1.** Assume the target density $p_{\text{target}}$ satisfies the following conditions:

(i) $p_{\text{target}}(\cdot) \in \mathcal{C}^2$. That is, it is second-order continuously differentiable;

(ii) *Log-Sobolev inequality*: there is a constant $\alpha > 0$ so that the following inequality holds for all continuously differentiable density $p$:

$$\text{KL}\big(p\|p_{\text{target}}\big) \leqslant \frac{1}{2\alpha} J_{p_{\text{target}}}(p). \tag{16}$$

(iii) $p_{\text{target}}$ is either compactly supported with $M := \|p_{\text{target}}\|_{L^\infty} < \infty$, or it decays sufficiently fast as $\|\mathbf{x}\|_2 \to \infty$:

$$\lim_{\|\mathbf{x}\|_2 \to \infty} \frac{p_{\text{target}}(\mathbf{x})}{\|\mathbf{x}\|_2^D} = 0.$$

**Assumption C.2.** Assume the target density $p_{\text{target}}$ satisfies the following additional conditions:

(i) There is a $L > 0$ so that for all $\mathbf{x}, \mathbf{y}$
$$\|p_{\text{target}}(\mathbf{x}) - p_{\text{target}}(\mathbf{y})\|_2^2 \leqslant L \|\mathbf{x} - \mathbf{y}\|_2^2 \quad \text{and} \quad \|\nabla p_{\text{target}}(\mathbf{x}) - \nabla p_{\text{target}}(\mathbf{y})\|_2^2 \leqslant L \|\mathbf{x} - \mathbf{y}\|_2^2.$$

### C.2    PROOFS OF THEOREM 4.1

*Proof.* Recall our dynamics is

$$d\mathbf{x}(t) = \nabla p_{\text{target}}(\mathbf{x}(t))\, dt + \sqrt{p_{\text{target}}(\mathbf{x}(t))}\, d\mathbf{w}(t).$$

The Fokker-Planck equation of our dynamics with density $p_t = p_t(\mathbf{x}) := p(\mathbf{x}, t)$ is

$$\partial_t p(\mathbf{x}, t) = \frac{1}{2} \nabla \cdot \Big( p_{\text{target}}(\mathbf{x}) p(\mathbf{x}, t) \nabla \log \frac{p(\mathbf{x}, t)}{p_{\text{target}}(\mathbf{x})} \Big). \tag{17}$$

This is due to the following derivation Risken & Risken (1996), where we demonstrated for the $D = 1$ case. The probability distribution $p(\mathbf{x}, t)$ of dynamics $\mathbf{x}(t)$ at point $\mathbf{x}$ and time $t$ with $f(x) = \partial_x p_{\text{target}}(x)$ and $g^2(x) = p_{\text{target}}(x)$ is governed by:

$$
\begin{aligned}
\frac{\partial p(x, t)}{\partial t} &= -\frac{\partial}{\partial x}\Big( f(x) p(x, t) \Big) + \frac{1}{2} \frac{\partial^2}{\partial x^2}\Big( g^2(x) p(x, t) \Big) \\
&= \frac{\partial}{\partial x}\Big( -\partial_x p_{\text{target}}(x) p(x, t) + \frac{1}{2} \frac{\partial}{\partial x}\Big( p_{\text{target}}(x) p(x, t) \Big) \Big) \\
&= \frac{1}{2} \frac{\partial}{\partial x}\Big( p_{\text{target}}(x) \partial_x p(x, t) - \partial_x p_{\text{target}}(x) p(x, t) \Big) \\
&= \frac{1}{2} \frac{\partial}{\partial x}\Big( p_{\text{target}}(x) p(x, t) \partial_x \log \frac{p(x, t)}{p_{\text{target}}(x)} \Big).
\end{aligned}
$$

Here, in the last equality we use the identity:

$$\partial_{\mathbf{x}} \log \frac{p(x,t)}{p_{\text{target}}(x)} = \frac{p_{\text{target}}(x)}{p(x,t)} \partial_{\mathbf{x}} \left( \frac{p(x,t)}{p_{\text{target}}(x)} \right)^2 = \frac{p_{\text{target}}(x) \partial_x p(x,t) - \partial_x p_{\text{target}}(x) p(x,t)}{p_{\text{target}}(x) p(x,t)}.$$

For a general $D$, the same computation can be carried out to derive Eq. (17).

We now prove the KL bound of convergence using a similar argument motivated by Vempala & Wibisono (2019).

$$\frac{\mathrm{d}}{\mathrm{d}t} \mathrm{KL}\big(p_t \| p_{\text{target}}\big)$$

$$= \frac{\mathrm{d}}{\mathrm{d}t} \int_{\mathbb{R}^D} p_t \log \frac{p_t}{p_{\text{target}}} \, \mathrm{d}\mathbf{x}$$

$$\overset{(a)}{=} \int_{\mathbb{R}^D} \frac{\partial}{\partial t} p_t \log \frac{p_t}{p_{\text{target}}} \, \mathrm{d}\mathbf{x} + \int_{\mathbb{R}^D} p_t \frac{\partial}{\partial t} \log \frac{p_t}{p_{\text{target}}} \, \mathrm{d}\mathbf{x}$$

$$\overset{(b)}{=} \int_{\mathbb{R}^D} \frac{\partial}{\partial t} p_t \log \frac{p_t}{p_{\text{target}}} \, \mathrm{d}\mathbf{x}$$

$$\overset{(c)}{=} \frac{1}{2} \int_{\mathbb{R}^D} \left[ \nabla_x \cdot \big( p_{\text{target}} p_t \nabla_x \log \frac{p_t}{p_{\text{target}}} \big) \right] \log \frac{p_t}{p_{\text{target}}} \, \mathrm{d}\mathbf{x}$$

$$\overset{(d)}{=} - \int_{\mathbb{R}^D} p_{\text{target}} p_t \left\| \nabla_x \log \frac{p_t}{p_{\text{target}}} \right\|^2 \, \mathrm{d}\mathbf{x}$$

$$\overset{(e)}{\leqslant} - M \int_{\mathbb{R}^D} p_t \left\| \nabla_x \log \frac{p_t}{p_{\text{target}}} \right\|^2 \, \mathrm{d}\mathbf{x}$$

$$= - M J_{p_{\text{target}}}(p_t)$$

$$\leqslant - \frac{M}{2\alpha} \mathrm{KL}\big(p_t \| p_{\text{target}}\big).$$

Here, (a) follows from the chain rule; (b) uses the identity $\int p_t \frac{\partial}{\partial t} \log \frac{p_t}{p_{\text{target}}} \, \mathrm{d}\mathbf{x} = \int \frac{\partial}{\partial t} p_t \, \mathrm{d}\mathbf{x} = \frac{\mathrm{d}}{\mathrm{d}t} \int p_t \, \mathrm{d}\mathbf{x} = 0$; (c) follows from the Fokker-Planck Eq. (17); (d) is due to integration by parts and Assumption C.1 (iii); and (e) comes from Assumption C.1 (iii).

Thus, applying Grönwall's inequality, we can get

$$\mathrm{KL}\big(p_t \| p_{\text{target}}\big) \lesssim e^{-2\alpha t} \mathrm{KL}\big(p_0 \| p_{\text{target}}\big).$$

Since $p_{\text{target}}$ satisfies the LSI, it also satisfies the Talagrand's inequality Otto & Villani (2000):

$$\frac{\alpha}{2} W_2^2\big(p_t, p_{\text{target}}\big) \leqslant \mathrm{KL}\big(p_t \| p_{\text{target}}\big).$$

Therefore, we have

$$W_2^2\big(p_t, p_{\text{target}}\big) \leqslant \frac{2}{\alpha} \mathrm{KL}\big(p_t \| p_{\text{target}}\big) \lesssim \frac{2}{\alpha} e^{-2\alpha t} \mathrm{KL}\big(p_0 \| p_{\text{target}}\big).$$

This completes the proof. We notice that "Talagrand's inequality implies concentration of measure of Gaussian type" allowing us to remove the compact support assumption on $p_{\text{target}}$ while maintaining the validity of the theorem. □

## C.3 PROOFS OF THEOREM 4.2

*Proof.* In the proof we will extensively using a simple form of Cauchy-Schwarz (CS) inequality:

$$(u_1 + u_2 + \cdots + u_n)^2 \leqslant n(u_1^2 + u_2^2 + \cdots + u_n^2),$$

for $u_i \in \mathbb{R}$, $i = 1, \cdots, n$. We aim at obtaining the following bound:

$$W_2^2(p_{T;\phi,\varphi}, p_{\text{target}}) \lesssim \varepsilon_{\text{est}}^2 T e^{LT} + \frac{2}{\alpha} e^{-\alpha T} \mathrm{KL}\big(p_0 \| p_{\text{target}}\big). \tag{18}$$

To achieve it, we compare the random vector processes $\{\mathbf{x}(t)\}_{t\in[0,T]}$ and $\{\widehat{\mathbf{x}}(t)\}_{t\in[0,T]}$, governed by the following dynamics:

$$\mathrm{d}\mathbf{x}(t) = \nabla p_{\text{target}}(\mathbf{x}(t))\,\mathrm{d}t + \sqrt{p_{\text{target}}(\mathbf{x}(t))}\,\mathrm{d}\boldsymbol{\omega}(t)$$

$$\mathrm{d}\widehat{\mathbf{x}}(t) = g_{\boldsymbol{\phi}}(\widehat{\mathbf{x}}(t))\,\mathrm{d}t + \sqrt{s_{\boldsymbol{\varphi}}(\widehat{\mathbf{x}}(t))}\,\mathrm{d}\widehat{\mathbf{w}}(t).$$

Their strong solutions in the Itô sense are:

$$\mathbf{x}(t) = \mathbf{x}(0) + \int_0^T \nabla p_{\text{target}}(\mathbf{x}(t))\,\mathrm{d}t + \int_0^T \sqrt{p_{\text{target}}(\mathbf{x}(t))}\,\mathrm{d}\boldsymbol{\omega}(t)$$

$$\widehat{\mathbf{x}}(t) = \widehat{\mathbf{x}}(0) + \int_0^T g_{\boldsymbol{\phi}}(\widehat{\mathbf{x}}(t))\,\mathrm{d}t + \int_0^T \sqrt{s_{\boldsymbol{\varphi}}(\widehat{\mathbf{x}}(t))}\,\mathrm{d}\widehat{\mathbf{w}}(t).$$

Set random vectors $\mathbf{a}(t) := \nabla p_{\text{target}}(\mathbf{x}(t)) - g_{\boldsymbol{\phi}}(\widehat{\mathbf{x}}(t))$ and $\mathbf{b}(t) := \sqrt{p_{\text{target}}(\mathbf{x}(t))} - \sqrt{s_{\boldsymbol{\varphi}}(\widehat{\mathbf{x}}(t))}$, we then have

$$\mathbb{E}\big[\,\|\mathbf{x}(T) - \widehat{\mathbf{x}}(T)\|_2^2\,\big] \leqslant \mathbb{E}\bigg[\Big(\mathbf{x}(0) - \widehat{\mathbf{x}}(0) + \int_0^T \mathbf{a}(t)\,\mathrm{d}t + \int_0^T \mathbf{b}(t)\,\mathrm{d}\boldsymbol{\omega}(t)\Big)^2\bigg]$$

$$\leqslant 3\mathbb{E}\big[\,\|\mathbf{x}(0) - \widehat{\mathbf{x}}(0)\|_2^2\,\big] + 3\mathbb{E}\Big[\big(\int_0^T \mathbf{a}(t)\,\mathrm{d}t\big)^2\Big] + 3\mathbb{E}\Big[\big(\int_0^T \mathbf{b}(t)\,\mathrm{d}\boldsymbol{\omega}(t)\big)^2\Big]$$

$$\lesssim \mathbb{E}\big[\,\|\mathbf{x}(0) - \widehat{\mathbf{x}}(0)\|_2^2\,\big] + T\mathbb{E}\Big[\int_0^T \mathbf{a}^2(t)\,\mathrm{d}t\Big] + \mathbb{E}\Big[\int_0^T \mathbf{b}^2(t)\,\mathrm{d}t\Big]$$

$$\lesssim \mathbb{E}\big[\,\|\mathbf{x}(0) - \widehat{\mathbf{x}}(0)\|_2^2\,\big] + T\mathbb{E}\Big[\int_0^T \|\nabla p_{\text{target}}(\mathbf{x}(t)) - \nabla p_{\text{target}}(\widehat{\mathbf{x}}(t))\|_2^2\,\mathrm{d}t\Big]$$

$$+ T\mathbb{E}\Big[\int_0^T \|\nabla p_{\text{target}}(\widehat{\mathbf{x}}(t)) - g_{\boldsymbol{\phi}}(\widehat{\mathbf{x}}(t))\|_2^2\,\mathrm{d}t\Big]$$

$$+ \mathbb{E}\Big[\int_0^T |p_{\text{target}}(\mathbf{x}(t) - p_{\text{target}}(\widehat{\mathbf{x}}(t))|\,\mathrm{d}t\Big] + \mathbb{E}\Big[\int_0^T |p_{\text{target}}(\widehat{\mathbf{x}}(t)) - s_{\boldsymbol{\varphi}}(\widehat{\mathbf{x}}(t))|\,\mathrm{d}t\Big]$$

$$\lesssim \mathbb{E}\big[\,\|\mathbf{x}(0) - \widehat{\mathbf{x}}(0)\|_2^2\,\big] + LT\int_0^T \mathbb{E}\big[\,\|\mathbf{x}(t) - \widehat{\mathbf{x}}(t)\|_2^2\,\big]\,\mathrm{d}t + \varepsilon_{\text{est}}^2 T.$$

Here, we apply the Cauchy-Schwarz (CS) inequality and the Itô isometry in the third inequality, the CS inequality and $(\sqrt{u} - \sqrt{v})^2 \leqslant |u - v|$ $(u, v \geqslant 0)$ in the fourth inequality, and the estimation error assumption in the last equality.

Since the dynamics in Eqs. (10) and (11) start from the same initial condition sampled from $p_0$, we have $\mathbb{E}\big[\,\|\mathbf{x}(0) - \widehat{\mathbf{x}}(0)\|_2^2\,\big] = 0$. Applying the Grönwall's inequality and the definition of the Wasserstein-2 distance, then we obtain

$$W_2^2(p_{T;\boldsymbol{\phi},\boldsymbol{\varphi}}, p_T) \lesssim \varepsilon_{\text{est}}^2 T e^{LT}.$$

Combining the above inequality and the result of Theorem 4.1 that

$$W_2^2(p_T, p_{\text{target}}) \lesssim \frac{2}{\alpha} e^{-\alpha T} \mathrm{KL}(p_0 \| p_{\text{target}}),$$

we finally derive the following inequality by applying CS inequality

$$W_2^2(p_{T;\boldsymbol{\phi},\boldsymbol{\varphi}}, p_{\text{target}}) \lesssim \varepsilon_{\text{est}}^2 T e^{LT} + \frac{2}{\alpha} e^{-\alpha T} \mathrm{KL}(p_0 \| p_{\text{target}}).$$

$\square$

# D  ALGORITHMS AND EXPERIMENTS WITH BELLMAN DIFFUSION

In Sec. D.1, we present the algorithms of Bellman Diffusion, highlighting its potential as a generative model. Sec. D.2 demonstrates the computational inefficiencies of naively applying existing DGMs to MDP tasks, further underscoring Bellman Diffusion's efficiency for such applications. Lastly, Sec. D.3 details the training configurations of Bellman Diffusion.

Figure 7: $15 \times 15$ randomly sampled images from our latent Bellman Diffusion model that is trained on the MNIST dataset. We can see that most of the results are high-quality.

| **Algorithm 3** Training | **Algorithm 4** Sampling |
|---|---|
| 1: **repeat** | 1: Sample $\mathbf{x}(0)$ from any initial distribution |
| 2: Sample real data: $\mathbf{x}_1, \mathbf{x}_2 \sim \mathcal{X}$ | 2: Set sampling steps $T$ |
| 3: Sample slice vectors: $\mathbf{v} \sim q(\mathbf{v}), \mathbf{w} \sim q(\mathbf{w})$ | 3: Set constant step size $\eta$ |
| 4: $\delta = \mathcal{N}(\mathbf{w}^\top \mathbf{x}_2 - \mathbf{w}^\top \mathbf{x}_1; 0, \epsilon)$ | 4: **for** $t = 0, 1, \ldots, T-1$ **do** |
| 5: $\bar{\mathcal{L}}_{\text{grad}}^{\text{slice}}(\phi; \epsilon) \approx (\mathbf{v}^\top \mathbf{g}_\phi(\mathbf{x}_1))^2 + \delta(\mathbf{v}^\top \nabla_{\mathbf{x}_1} \mathbf{g}_\phi(\mathbf{x}_1)\mathbf{v})$ | 5: $\mathbf{z} \sim \mathcal{N}(\mathbf{0}, \mathbf{I}_D)$ |
| 6: $\bar{\mathcal{L}}_{\text{id}}^{\text{slice}}(\varphi; \epsilon) \approx s_\varphi(\mathbf{x}_1)^2 - 2\delta s_\varphi(\mathbf{x}_1)$ | 6: $\Delta = \mathbf{g}_\phi(\mathbf{x}(\eta t))\eta + \sqrt{s_\varphi(\mathbf{x}(\eta t))\eta}\mathbf{z}$ |
| 7: Update parameter $\phi$ w.r.t. $-\nabla_\phi \bar{\mathcal{L}}_{\text{grad}}^{\text{slice}}(\phi; \epsilon)$ | 7: $\mathbf{x}(\eta(t+1)) = \mathbf{x}(\eta t) + \Delta$ |
| 8: Update parameter $\varphi$ w.r.t. $-\nabla_\varphi \bar{\mathcal{L}}_{\text{grad}}^{\text{slice}}(\varphi; \epsilon)$ | 8: **end for** |
| 9: **until** converged | 9: **return** $\mathbf{x}(\epsilon T)$ |

## D.1 BELLMAN DIFFUSION'S TRAINING AND SAMPLING AS A DGM

In this section, we detail the algorithms for training (Alg. 3) and sampling (Alg. 4) in Bellman Diffusion as a general DGM.

## D.2 DISFAVORED FULL TRAJECTORY SAMPLING

As mentioned in Sec. 2, a DGM that is qualified to be applied with the efficient Bellman update needs to satisfy some linearity condition, otherwise one can only sample full state-action trajectories to train the DGM, which is too costly for many RL environments. To understand this point, suppose that there is an 1-dimensional maze with $N$ blocks, with a robot moving from the leftmost block to the rightmost block. If one directly trains the return model with the returns computed from full trajectories, then the robot has to try to move to the final block after each action, resulting in a time complexity at least as $\mathcal{O}(N \cdot N) = \mathcal{O}(N^2)$ for every episode. In contrast, if the return model can be trained with partial trajectories (e.g., 1 step) through the Bellman equation, then the time complexity would be significantly reduced (e.g., $\mathcal{O}(N^2)$). There are many RL environments where the number

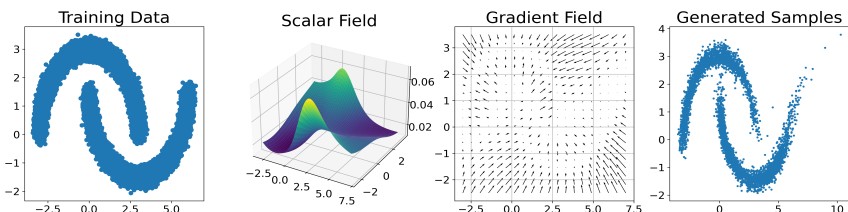

Figure 8: *Bellman Diffusion learns unusually clustered data.* The subfigures, from left to right, show the training data, estimated density field, gradient field, and generated samples.

$N$ can be very big. For example, StarCraft II (Vinyals et al., 2017) and Counter-Strike (Pearce & Zhu, 2022), where a full trajectory can contain over ten thousand steps.

### D.3 EXPERIMENT SETTINGS

Unless specified, we construct the gradient and scalar field models $\mathbf{g}_\phi(\mathbf{x})$ and $s_\varphi(\mathbf{x})$ using MLPs (Pinkus, 1999). We employ Adam (Kingma, 2014) for optimization, without weight decay or dropout. The parameter $\epsilon$ in the loss functions $\bar{\mathcal{L}}_{\mathrm{grad}}^{\mathrm{slice}}(\phi; \epsilon)$ and $\bar{\mathcal{L}}_{\mathrm{id}}^{\mathrm{slice}}(\varphi; \epsilon)$ ranges from 0.1 to 1.0, depending on the task. For the sampling dynamics defined in Eq. (10), we typically set $T = 300$ and $\eta = 0.1$. All models are trained on a single A100 GPU with 40GB memory, taking only a few tens of minutes to a few hours.

## E ADDITIONAL EXPERIMENTS

Due to the limited space, we put the minor experiments here in the appendix. The main experiments involving field estimation, generative modeling, and RL are placed in the main text.

### E.1 SYNTHETIC DATA GENERATION

**2-dimensional moon-shaped data.** To demonstrate the ability of Bellman Diffusion to learn distributions with disjoint supports, we test it on the two moon dataset, where samples cluster into two disjoint half-cycles, as shown in the leftmost subfigure of Fig. 8.

The right three parts of Fig. 8 shows that the estimated scalar and gradient fields $p_{\mathrm{target}}(\mathbf{x})$ and $\nabla p_{\mathrm{target}}(\mathbf{x})$ match the training samples, with correctly positioned density peaks (leftmost subfigure) and critical points (middle subfigure). Our diffusion sampling dynamics accurately recover the shape of the training data, even in low-density regions. Thus, we conclude that Bellman Diffusion is effective in learning from complex data.

**Comparison of Bellman Diffusion and DDPM on 2-dimensional MoG.** We provide an additional comparison of generated samples from DDPM and Bellman Diffusion on a MoG dataset with three modes. As the setup in Fig. 5, the training distribution consists of three modes with weights of 0.45, 0.45, and 0.1. The results are shown in Fig. 9. We observe that the generated samples from DDPM (right subfigure) fail to capture the different weights of these modes. In contrast, Bellman Diffusion (left subfigure) successfully recovers the three modes with their respective weights, as also demonstrated in Fig. 5.

The reason Bellman Diffusion may learn different modes of $p_{\mathrm{target}}$ is that our training objectives directly model $s_\varphi \approx p_{\mathrm{target}}$ (and its gradient, $\mathbf{g}_\phi \approx \nabla p_{\mathrm{target}}$). As a result, it can learn different modes within $p_{\mathrm{target}}$. This contrasts with diffusion models, which learn the score function $\nabla \log p_{\mathrm{target}}$ for generation.

To illustrate this difference, consider an example where $p_{\mathrm{target}} = a p_{\mathrm{target}}^{(1)} + b p_{\mathrm{target}}^{(2)}$, which represents a mixture of two modes with weights $a$ and $b$, and where the supports of $p_{\mathrm{target}}^{(1)}$ and $p_{\mathrm{target}}^{(2)}$ are disjoint.

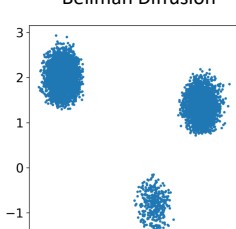
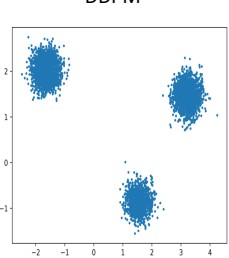

Figure 9: Comparison of generated samples from MoG with three modes between Bellman Diffusion and DDPM. (Left) Bellman Diffusion accurately captures the three modes with different weights. (Right) DDPM struggles to reflect the correct weight distribution of the target.

| Method | Abalone | Telemonitoring |
|--------|---------|----------------|
| Bellman Diffusion w/ $\epsilon = 0.5, n = 1$ | 0.975 | 2.167 |
| Bellman Diffusion w/ $\epsilon = 1.0, n = 1$ | 1.113 | 2.379 |
| Bellman Diffusion w/ $\epsilon = 0.1, n = 1$ | 0.875 | 2.075 |
| Bellman Diffusion w/ $\epsilon = 0.01, n = 1$ | 1.567 | 3.231 |
| Bellman Diffusion w/ $\epsilon = 0.5, n = 2$ | 0.912 | 2.073 |
| Bellman Diffusion w/ $\epsilon = 0.5, n = 3$ | 0.895 | 1.951 |

Table 1: The experiment results of our case studies, with Wasserstein distance as the metric.

For a point $\mathbf{x}$ in the support of $p_{\text{target}}^{(1)}$, we have:

$$\nabla_{\mathbf{x}} \log p_{\text{target}}(\mathbf{x}) = \nabla_{\mathbf{x}} \log a + \nabla_{\mathbf{x}} \log p_{\text{target}}^{(1)}(\mathbf{x}) = \nabla_{\mathbf{x}} \log p_{\text{target}}^{(1)}(\mathbf{x}).$$

Similarly, for a point $\mathbf{x}$ in the support of $p_{\text{target}}^{(2)}$, we have:

$$\nabla_{\mathbf{x}} \log p_{\text{target}}(\mathbf{x}) = \nabla_{\mathbf{x}} \log p_{\text{target}}^{(2)}(\mathbf{x}).$$

This example illustrates that, by using the score function (as in the case of diffusion models), we are unable to recover the weights $a$ and $b$ of the mixture components.

### E.2 IMAGE GENERATION

While image generation is not the main focus of our paper, we show that Bellman Diffusion is also promising in that direction. We adopt a variant of the widely used architecture of latent diffusion (Rombach et al., 2022), with VAE to encode images into latent representations and Bellman Diffusion to learn the distribution of such representations. We run such a model on MNIST (Deng, 2012), a classical image dataset. The results are shown in Fig. 7. We can see that most generated images are high-quality. This experiment verify that Bellman Diffusion is applicable to high-dimensional data, including image generation.

### E.3 ABLATION STUDIES

There are some important hyper-parameters of Bellman Diffusion that need careful studies to determine their proper values for use. This part aims to achieve this goal. We adopt two tabular datasets: Abalone and Telemonitoring, with the Wasserstein distance as the metric.

**The variance of Gaussian coefficients.** The loss functions $\bar{\mathcal{L}}_{\text{grad}}(\phi; \epsilon), \bar{\mathcal{L}}_{\text{id}}(\varphi; \epsilon)$ of both gradient and scalar matching contain a term $\epsilon$, which is to relax their original limit forms for practical computation. As shown in the first 4 rows of Table 1, either too big or too small value of term $\epsilon$ leads to worse performance of our Bellman Diffusion model. These experiment results also make sense because too big $\epsilon$ will significantly deviate the loss functions from their limit values, and too small $\epsilon$ will also cause numerical instability.

**Number of slice vectors.** Intuitively, more slice vectors will make our loss estimation more accurate, leading to better model performance. The experiment results in the first and the last two rows of Table 1 confirm this intuition, but also indicate that such performance gains are not notable. Therefore, we adopt $n = 1$ slice vectors in experiments to maintain high efficiency.

