# OpenReview forum: "Bellman Diffusion: Generative Modeling as Learning a Linear Operator in the Distribution Space"
_ICLR.cc/2025/Conference — Submitted to ICLR 2025_

### Official Review · Reviewer_Duad · 2024-11-03

**Soundness:** 2
**Presentation:** 2
**Contribution:** 3
**Rating:** 5
**Confidence:** 4

**Summary:**

This paper introduces Bellman Diffusion, a deep generative modeling framework designed to overcome limitations in applying existing generative models to Markov Decision Processes (MDPs), especially when the downstream task is distributional Reinforcement Learning (RL). The authors identify a key issue: the nonlinearity of modern deep generative models conflicts with the linearity required by the Bellman equation in MDPs. To address this, they propose modeling the gradient field and scalar field of the target distribution directly, which preserves linearity.

**Strengths:**

1.	The paper put forth an innovative solution to a well-defined and very interesting problem in applying generative models to MDPs.

2.	The authors provide a rigorous theoretical basis for their algorithmic innovations, including convergence guarantees and error bound analysis.

3.	The paper offers detailed algorithms for training and sampling, making the method easier to follow for implementation purposes.

4.	The authors demonstrate the effectiveness of Bellman Diffusion on both synthetic and real datasets, and also show improvements towards distributional RL tasks.

**Weaknesses:**

1.	The method introduces additional complexity compared to traditional histogram-based approaches, which may limit its adoption in some practical scenarios.

2.	While the paper shows improvements over histogram-based methods, comparisons with other advanced generative modeling techniques in RL contexts are not extensively explored.

3.	The paper does not thoroughly address the scalability of the method to very high-dimensional problems or large-scale RL environments.

4.	The authors must improve quality and readability of their figures and other visualizations. For instance, Figure 4 is especially hard to follow.

5. There are multiple typos/grammatical errors across the manuscript which should be fixed.

**Questions:**

In addition to the aforementioned weakness comments, I request the authors to provide clarifications for the following questions.

1.	Are there any limitations or failure cases where Bellman Diffusion does not perform well compared to other generative modeling approaches?

2.	How computationally expensive is the training process for Bellman Diffusion compared to other generative modeling approaches? Are there any tricks used to improve training efficiency?

3.	The paper shows experimental anecdotes on how Bellman Diffusion could effectively learn multi-modal distributions. How does it compare to other methods like normalizing flows or mixture density networks for multi-modal density estimation?

4.	The slice trick is used to improve sample efficiency. How sensitive is the method to the choice of slice vector distributions q(v) and q(w)? Are there guidelines for selecting these?

5.	Can the authors please share details of the implementation codebase that was used for the experimental evaluations ?

---

> ### Author Response · Authors · 2024-11-24
> **Rebuttal, Section 1**
>
> We thank the reviewer for such comprehensive and constructive feedback.
>
> > ### [***Weakness-1***] The method introduces additional complexity compared to traditional histogram-based approaches.
>
> Thank you for your thoughtful comment. We acknowledge the concern about the added complexity compared to traditional histogram-based approaches. However, we would like to emphasize the following points to clarify the situation:
>
> **[Similar Complexity in Core Computation]** Both histogram-based approaches and Bellman Diffusion are fundamentally based on the Bellman update equation. The primary difference lies in how the distribution of returns is learned at each state. As a result, the overall computational complexity remains comparable. In fact, histogram-based approaches often require additional computational overhead when handling large numbers of bins, as the method involves iterating over $N$ bins to estimate the distribution. If $N$ is large, this can increase the computational cost.
>
> **[Discretization Errors in Histogram-Based Methods]** Histogram-based methods inherently suffer from discretization errors when approximating a continuous return distribution using a finite number of bins. These errors can accumulate over multiple Bellman updates, leading to instability in the learned value function, as reflected in the fluctuating returns in Figure 6.
>
> **[Why Bellman Diffusion?]**  In contrast, Bellman Diffusion is a principled method for learning continuous distributions, leveraging the linearity property to simplify training (as shown in Algorithms 1 and 3), while remaining compatible with the Bellman update. This results in a smoother and more accurate approximation of the return distribution. It also demonstrates faster and more stable convergence compared to histogram-based approaches, as illustrated in Figure 6. We would appreciate it if the reviewer could highlight specific aspects of Bellman Diffusion that may add complexity during training.
>
> > ### [***Weakness-2***] Comparisons with other advanced generative modeling techniques in RL contexts are not extensively explored.
>
> We appreciate the reviewer’s comment. To the best of our knowledge, Bellman Diffusion is the first deep generative model (DGM) that efficiently and continuously (not discretely) models return distribution at each state while being compatible with the Bellman update. This efficiency stems from the linearity property discussed in Section 2. Our analysis shows that none of the existing DGMs satisfy this linearity, which, if naively applied to model return distributions for the Bellman update, would lead to significantly higher computational costs.
>
> As detailed in Appendix D.2, a DGM suitable for efficient Bellman updates must satisfy a linearity condition; otherwise, training would require full state-action trajectories, which is prohibitively expensive in many RL environments. For example, in a 1-dimensional maze with $N$ blocks, directly training the return model with full trajectories would incur a time complexity of $\mathcal{O}(N^2)$ per episode. In contrast, using partial trajectories through the Bellman equation drastically reduces the time complexity. With environments like CartPole in Fig. 5, where $N = 500$, the complexity would be at least $\mathcal{O}(500^2)$, which is computationally expensive.
>
> Thus, we believe comparing Bellman Diffusion to other DGMs is not reasonable, given the significantly higher computational demands of those models. Indeed, establishing a new DGM that is compatible with Bellman update with cheap computation cost is one of our motivations.
>
> > ### [***Weakness-3***] The scalability of the method to very high-dimensional problems or large-scale RL environments.
>
> **[Scalability of Dimensionality]** We thank the reviewer for raising the important concern regarding the scalability of Bellman Diffusion to high-dimensional problems and large-scale RL environments. We would like to clarify that our primary focus in this paper is on bridging deep generative models (DGMs) to MDPs and distributional RL, specifically replacing the classic histogram-based approximation of the return distribution with our Bellman Diffusion method, which directly learns a continuous return distribution. Since the return distribution is typically one-dimensional in this context, scalability to high-dimensional data is not the central challenge addressed in this work. However, to provide evidence that Bellman Diffusion can be scaled to higher-dimensional environments, we included preliminary proof-of-concept results in Table 6 and Figure 7, with further details in the (original) manuscript.

---

> ### Author Response · Authors · 2024-11-24
> **Rebuttal, Section 2**
>
> **[More Complicated RL Environments]** In principle, Bellman Diffusion is designed to be a generic method, and we do not anticipate any fundamental limitations that would prevent its application to more complex RL environments. We believe that the superior stability and faster convergence of our method, stemming from its ability to directly learn the continuous return distribution, would translate well to more intricate environments. This is in contrast to methods like C51, which suffer from discretization errors at each state that accumulate and propagate through state transitions, limiting their scalability.
>
> To address this, we are actively working on applying Bellman Diffusion to Atari environments from OpenAI’s Gym library, which feature more complex dynamics and larger state spaces than those used in the manuscript. While we are making every effort to complete these experiments during the discussion period, we require additional time to adapt our codebase to accommodate the unique aspects of RL environments, such as reward shaping, environment interfaces, and visual input processing. This is more involved than the typical adjustments needed for supervised learning, where the primary change is the dataset. Given these challenges and the limited timeframe, we may not be able to finalize the results in time for the rebuttal. However, if these results are not included in the rebuttal, we commit to incorporating them in the final manuscript to further demonstrate the scalability and applicability of Bellman Diffusion to more complex RL tasks.
>
> > ### [***Weakness-4,5***] The authors must improve quality and readability of their figures and other visualizations. Multiple typos/grammatical errors
>
> We sincerely thank the reviewer for pointing out these issues. In response, we have carefully revised the figures, to improve their clarity and readability. We have also reviewed the entire manuscript thoroughly and corrected the typos and grammatical errors we identified. We deeply appreciate the reviewer’s efforts and welcome any additional feedback on specific areas we might have missed.
>
> > ### [***Question-1***] Any limitations or failure cases where Bellman Diffusion does not perform well compared to other generative modeling approaches?
>
> Thank you for your insightful question. We would like to address the comparison of Bellman Diffusion with other generative modeling approaches. When applied to Markov Decision Processes (MDP), as mentioned in our response to W2, one key motivation of Bellman Diffusion is to establish a new DGM compatible with the Bellman update while maintaining low computational cost. Therefore, comparing Bellman Diffusion with DGMs that lack linearity and incur significant computational overhead, as highlighted in Section 2 and Appendix D.1, is not reasonable.
>
> For pure generation tasks, Bellman Diffusion generation process involves solving a Stochastic Differential Equation (SDE) (as shown in Eq. (13) of the original manuscript), which may result in slower generation speeds when compared to one-step generators such as Variational Autoencoders (VAE) or Generative Adversarial Networks (GAN). This is a limitation that Bellman Diffusion shares with diffusion models, whose sampling also involves solving differential equations. However, we see this as an area for potential future research. Much like the advancements in fast sampling for Diffusion Models, we believe that more sophisticated SDE solvers (e.g., [1, 2]) or techniques such as distilling the multi-step sampling process of a pre-trained Bellman Diffusion into a one-step generation (e.g., [3]) could significantly improve the efficiency of Bellman Diffusion's generation process.
>
> It is important to note, however, that in our experiments on distributional RL, where Bellman Diffusion is used to model scalar return distributions, the sampling process is relatively fast because it only involves one-dimensional SDE solving, making it suitable for these specific tasks.
>
> **References**
>
> [1] Lu, C., Zhou, Y., Bao, F., Chen, J., Li, C., & Zhu, J. (2022). Dpm-solver: A fast ode solver for diffusion probabilistic model sampling in around 10 steps. Advances in Neural Information Processing Systems, 35, 5775-5787.
>
> [2] Gonzalez, M., Fernandez Pinto, N., Tran, T., Hajri, H., & Masmoudi, N. (2024). Seeds: Exponential sde solvers for fast high-quality sampling from diffusion models. Advances in Neural Information Processing Systems, 36.
>
> [3] Song, Y., Dhariwal, P., Chen, M., & Sutskever, I. (2023). Consistency models. arXiv preprint arXiv:2303.01469.

---

> ### Author Response · Authors · 2024-11-24
> **Rebuttal, Section 3**
>
> ### [***Question-2***] How computationally expensive is the Bellman Diffusion compared to other generative models? Are any tricks used to improve training efficiency?
>
> As mentioned in our response to Weakness-2, we emphasize that one key motivation of Bellman Diffusion is to establish a new DGM compatible with the Bellman update while maintaining low computational cost. Comparing it to DGMs lacking linearity is unreasonable, given the significant computational overhead of these models, as discussed in Section 2 and Appendix D.1.
>
> Regarding training efficiency, we observe that the computational cost of training Bellman Diffusion is not prohibitive. The training process is stable and straightforward, following Algorithm 1 and Algorithm 3, without requiring additional tricks or optimizations. While we acknowledge a potential area for future research, in our current implementation, two separate networks are used to learn $s_{\varphi} \approx p_{target}$ and its gradient $\nabla p$. We believe that sharing weights between these networks could reduce computation, further improving training efficiency.
>
> ### [***Question-3***] How does Bellman Diffusion compare to other methods like normalizing flows or mixture density networks for multi-modal density estimation?
>
> As our training objectives directly learn $s_{\varphi} \approx p$ (and its gradient $\nabla p$). This may allow Bellman Diffusion to learn different modes of the data distribution $p$. For reference, in our reply to W2 of Reviewer 57hc, we also provide a detailed comparison between Bellman Diffusion and Score-based Diffusion Models, with a theoretical derivation to show that the latter cannot recognize the data imbalance of mixture distributions. Additionally, we empirically showed in the new Fig. 9 of our paper that Bellman Diffusion successfully recognized the data imbalance and vanilla diffusion models failed to do so.
>
> In principle, we believe that normalizing flows or mixture density networks could also learn different modes of $p_{\mathrm{target}}$ [1]. However, since they do not satisfy the linearity property defined in Sec. 2, their applications to MDPs may be limited.
>
> [1] Normalizing flows for probabilistic modeling and inference. Journal of Machine Learning Research, 22(57), 1-64.
>
> ### [***Question-4***]  The slice trick is used to improve sample efficiency. How sensitive is the method to the choice of slice vector distributions q(v) and q(w)? Are there guidelines for selecting these?
>
> The use of slice vectors is inspired by Hutchinson’s trace estimator [1], which estimates the trace of a matrix $\mathbf{A} \in \mathbb{R}^{D \times D}$ via
> $$
> \mathrm{Trace}(\mathbf{A})=E_{q(\mathbf{v})}[\mathbf{v}^\top \mathbf{A} \mathbf{v}] \approx \frac{1}{M} \sum_{i=1}^M \mathbf{v}_i^\top \mathbf{A} \mathbf{v}_i.
> $$
> Here, $M$ is the number of slice vectors, and $\mathbf{v} \in \mathbb{R}^D$ is a random vector such that $\mathbb{E}[\mathbf{v} \mathbf{v}^\top] = \mathbf{I}$. In principle, any distribution $q(\mathbf{v})$ for the slice vector such that $\mathbb{E}[\mathbf{v} \mathbf{v}^\top] = \mathbf{I}$ can be valid for this purpose.
>
> However, prior works [2, 3, 4] typically consider the Rademacher distribution (the uniform distribution over $\{+1, -1\}^D$) or the multivariate standard normal distribution $\mathcal{N}(\mathbf{0}, \mathbf{I}_D)$ as popular choices for slice vectors.
>
> In our implementation, we tested both the Rademacher distribution and the standard normal distribution. Our empirical results demonstrate that the method is not particularly sensitive to these choices, as shown in [4].
>
> **References**
>
> [1] A stochastic estimator of the trace of the influence matrix for Laplacian smoothing splines. Communications in Statistics-Simulation and Computation, 18(3), 1059-1076.
>
> [2]  Modern analysis of hutchinson's trace estimator. In 2021 55th Annual Conference on Information Sciences and Systems (CISS) (pp. 1-5). IEEE.
>
> [3] Hutch++: Optimal stochastic trace estimation. In Symposium on Simplicity in Algorithms (SOSA) (pp. 142-155). Society for Industrial and Applied Mathematics.
>
> [4] Sliced score matching: A scalable approach to density and score estimation. In Uncertainty in Artificial Intelligence (pp. 574-584). PMLR.
>
> ### [***Question-5***] Can the authors please share details of the implementation codebase that was used for the experimental evaluations?
>
> To support reproducibility during the review stage, we provide an initial codebase. Once the paper is accepted, we will share the polished and finalized version of the code.

---

### Official Review · Reviewer_57hc · 2024-11-04

**Soundness:** 3
**Presentation:** 2
**Contribution:** 2
**Rating:** 5
**Confidence:** 3

**Summary:**

This paper introduces Bellman Diffusion, a novel generative model designed to approximate both the gradient and scalar field of the data’s probability distribution. Bellman Diffusion is distinguished by its linear modeling property, meaning that if we aim to model a combined distribution f+g, we can simply sum the fields of f and g. This property is particularly advantageous for distributional reinforcement learning (RL). By first modeling the distribution of returns in terminal states, we can subsequently perform Bellman-like updates on the approximated fields, ultimately deriving the distribution for all states in the Markov Decision Process (MDP). In essence, Bellman Diffusion’s linearity makes it well-suited for distribution modeling tasks where data exhibits a linear structure. The authors validate Bellman Diffusion’s effectiveness through evaluations on low-dimensional toy tasks, high-dimensional benchmarks, and applications in distributional RL.

**Strengths:**

The motivation behind this paper is both clear and compelling. In reinforcement learning, many distributions exhibit inherent structures that can be effectively described through Bellman iteration. Beyond the return or value distribution, other examples include the successor state distribution, which describes the probability of transitioning to a state s' at any time after taking an action a in a state s. Learning these distributions—even with powerful generative models—can be challenging if the data’s inherent structure isn’t properly respected. In this regard, Bellman Diffusion offers a practical solution, enabling the modeling of such structured distributions in a way that aligns with their underlying characteristics.

**Weaknesses:**

The presentation of the paper is poor. Since the authors motivate their approach through its application in distributional reinforcement learning, it would be more effective to explain how Bellman Diffusion can be integrated into and facilitate distributional RL (currently in Appendix D) directly within the main method section.

The evaluation of the proposed method is quite limited in both comparisons with existing methods and also the explanations about the performance.
+ In Section 6.1, the authors primarily demonstrate that Bellman Diffusion can model synthetic data distributions (though there are apparent modeling inconsistencies). Including comparisons with established baseline methods, such as DDPM, would provide a clearer picture of the advantages Bellman Diffusion offers.
+ In Section 6.2, given that the tasks lack internal structures, it raises the question of why Bellman Diffusion outperforms DDPM in certain tasks. Further explanation here would clarify the observed performance benefits and also certain claims made in line 431. Regarding Section 6.3, the benchmark environments (FrozenLake and CartPole) are too simple to truly assess the method’s effectiveness. These environments have relatively small state and action spaces, leading to straightforward return distributions. In contrast, existing methods like C51 are validated on more complex tasks, such as those in Atari environments, which exhibit greater structural complexity. In light of this, I think it would be necessary to extend the evaluations to more complex environments to truly reflect the effectiveness of the proposed method.

**Questions:**

One concurrent work [1] also seems to optimize the diffusion model in a temporal difference manner, and they used existing diffusion models, rather than proposing a new SDE with linear operators as in Bellman Diffusion. Although I understand that [1] deals with the successor measure instead of value distribution, could the authors briefly discuss about the difference or relationship between [1] and Bellman Diffusion?

[1] Liam Schramm and Abdeslam Boularias. Bellman Diffusion Models.

---

> ### Author Response · Authors · 2024-11-24
> **Rebuttal, Section 1**
>
> We thank the reviewer for such comprehensive and constructive feedback.
>
> > ### [***Weakness-1***] The presentation of the paper is poor. It would be more effective to explain Bellman Diffusion within the main method section.
>
> To elaborate on how Bellman Diffusion is motivated and applied to MDPs, particularly in distributional RL tasks, we have moved and elaborated the contents originally in Appendix D.2 to the current Section 5.
>
> Thank you for the valuable comments and careful check. If there are additional unclear points, please kindly let us know, and we will be happy to incorporate your advice to further improve our paper.
>
> > ### [***Weakness-2***]  The evaluation of the proposed method is quite limited in both comparisons with existing methods and also the explanations about the performance.
>
> **About Modeling Synthetic Data:** We appreciate the reviewer’s constructive comment. In Appendix E.1 of our revised manuscript, we provide an additional comparison of generated samples from DDPM and Bellman Diffusion on a Mixture of Gaussians (MoG) dataset with three modes. The training distribution consists of three modes with weights of 0.45, 0.45, and 0.1. However, the generated samples from DDPM fail to capture the different weights of these modes (please refer to our new Fig. 9 in the appendix). In contrast, Bellman Diffusion successfully recovers the three modes with their respective weights, as also demonstrated in Fig. 5 of the manuscript. We discuss the potential reasons for this in the following reply.
>
> **Why Bellman Diffusion is Capable in Certain Cases?** The reason Bellman Diffusion may learn different modes of $p_{target}$ is that our training objectives directly model $s_{\varphi} \approx p_{target}$ (and the gradient field). As a result, it can learn different modes within $p_{target}$. This contrasts with diffusion models, which learn the score function $\nabla \log p_{target}$ for generation.
>
> To illustrate this difference, consider an example where $p_{\mathrm{target}} = a p_{\mathrm{target}}^{(1)} + b p_{\mathrm{target}}^{(2)}$, which represents a mixture of two modes with weights $a$ and $b$, and where the supports of $p_{\mathrm{target}}^{(1)}$ and $p_{\mathrm{target}}^{(2)}$ are disjoint.
>
> For a point $\mathbf{x}$ in the support of $p_{\mathrm{target}}^{(1)}$, we have:
> $$\nabla_{\mathbf{x}} \log p_{\mathrm{target}}(\mathbf{x}) = \nabla_{\mathbf{x}} \log a + \nabla_{\mathbf{x}} \log p_{\mathrm{target}}^{(1)}(\mathbf{x}) = \nabla_{\mathbf{x}} \log p_{\mathrm{target}}^{(1)}(\mathbf{x}).$$
> Similarly, for a point $\mathbf{x}$ in the support of $p_{\mathrm{target}}^{(2)}$, we have:
> $$
> \nabla_{\mathbf{x}} \log p_{\mathrm{target}}(\mathbf{x}) = \nabla_{\mathbf{x}} \log p_{\mathrm{target}}^{(2)}(\mathbf{x}).
> $$
> This example illustrates that, by using the score function (as in the case of diffusion models), current diffusion-based generative baselines are unable to recognize the weights $a$ and $b$ of the mixture components.
>
> In contrast, Bellman Diffusion, which directly models the target distribution $p_{\mathrm{target}}$ and its gradient, is able to learn and capture these modes more effectively. This may explain the empirical results in Sec. 6.2.
>
> **More Complicated Environments:** We thank the reviewer for this insightful comment. In principle, Bellman Diffusion is designed to be a generic framework, and we do not foresee any inherent limitations to its application in more complex RL environments. Based on our understanding, the method’s superior stability and faster convergence should extend to these scenarios, as it learns a continuous return distribution that inherently avoids the discretization errors in the C51 baseline, which accumulate and propagate through state transitions.
>
> To address this, we are currently working to implement Bellman Diffusion in Atari environments from OpenAI’s Gym library, which offer more intricate dynamics and state spaces than the examples provided in the manuscript. While we are striving to complete these experiments during the discussion period, the complexity of fitting a codebase to a new environment (e.g., interaction interface and reward reshaping) and the limited timeframe may not allow us to finalize the results. If not included in the rebuttal, we commit to providing these results in the final manuscript to further demonstrate the method’s scalability and applicability to more challenging RL tasks.

---

> ### Author Response · Authors · 2024-11-24
> **Rebuttal, Section 2**
>
> > ### [***Question-1***] To compare with a possibly concurrent work
>
> We thank the reviewer for pointing out this concurrent work. While Bellman Diffusion Models (BDM) and our proposed method share some conceptual connections, they address different problem settings and have distinct goals:
>
> **[Different RL Paradigms]** BDM is grounded in standard value-based RL and aims to estimate the *successor state measure (SSM)*. In contrast, our work focuses on *distributional RL*, where the goal is to model the entire return distribution for each state, rather than just its first moment (i.e., the value). This difference reflects a fundamental distinction in the types of information each method seeks to capture.
>
> **[Policy Derivation]** While BDM estimates the SSM, it does not explicitly discuss how a policy can be derived from it. In comparison, distributional RL, as utilized in our method, provides a direct framework for deriving a feasible policy from the return distribution. This makes our approach more readily applicable to practical RL tasks.
>
> **[Methodological and Experimental Contributions]** BDM focuses primarily on introducing the concept of SSM estimation using diffusion models, but it does not include experimental validation to support its methodology. In contrast, we provide a comprehensive framework with experimental results that demonstrate the effectiveness of Bellman Diffusion in both RL tasks and generative modeling.
>
> While BDM offers valuable insights into the use of diffusion models in RL, our work addresses a complementary yet distinct problem. To clarify these differences, we have included a brief discussion in the revised manuscript. We thank the reviewer for encouraging us to draw this comparison.

---

> > ### Comment · Reviewer_57hc · 2024-11-25
> >
> > Thanks, I will check the results on Atari if they can be updated before the deadline of the discussion period.
> >
> > + **About: Why Bellman Diffusion is Capable in Certain Cases?**
> >
> >   Thanks for your further explanations about the possible advantages of modeling probabilities rather than the score functions. However, as Diffusion Models perturb clean data with different levels of Gaussian noises, the key assumption that $p_{\text{target}}^{(1)}$ is disjoint from $p_{\text{target}}^{(2)}$ may not hold. The advantages of Bellman Diffusion over Diffusion Models in realistic tasks (such as image generation) still remain unclear to me.
> >
> > + **About: Comparison to BDM**
> >
> >   I understand that the target of BDM is to approximate the successor measure rather than the return distribution. However, I would like to note that the objective in BDM (the first equation in Section 3), is nearly identical to Bellman Diffusion if you translate $d^\pi$ into the return distribution and $T$ into the immediate reward distribution. Therefore, I expect that the techniques used in BDM can be transferred to modeling the return distribution. A key advantage of BDM is that it can directly use DDPM, rather than using a new SDE and use slicing tricks to train the model. Could the authors comment on this?

---

> > > ### Author Response · Authors · 2024-11-27
> > >
> > > We appreciate your comments and the opportunity to provide further clarification.
> > >
> > > **[Clarification About Our Claim]** While applying varying levels of Gaussian noise in Diffusion Models may mitigate certain challenges, our empirical findings indicate that these models still struggle to effectively recover modes with distinct weightings. Additionally, no straightforward theoretical framework supports their ability to achieve this recovery.
> > >
> > > We would like to emphasize that the primary objective of Bellman Diffusion is not to claim superiority over Diffusion Models in tasks such as image generation. Instead, Bellman Diffusion is designed to leverage its inherent compatibility with the Bellman update, a property rooted in the linearity of its formulation. This compatibility enables Bellman Diffusion to excel in scenarios where modeling the density function and its gradient directly is advantageous.
> > > This direct modeling allows Bellman Diffusion to naturally recover weighted modes of the target distribution, a capability that is not as straightforward in Diffusion Models.
> > >
> > > **[About BDM]** The successor measure is defined over the state space, hugely different from the return distribution.
> > > In a broad sense, that measure even comes from value-based RL. We did not see the possibility that BDM can be easily applied to distributional RL.
> > > We would appreciate it if the reviewer could provide strict mathematics or codes verifying such possibility.

---

### Official Review · Reviewer_sSvE · 2024-11-04

**Soundness:** 4
**Presentation:** 3
**Contribution:** 3
**Rating:** 6
**Confidence:** 2

**Summary:**

This paper proposes a new generative framework, Bellman Diffusion, tailored for applications in Markov Decision Processes (MDPs) and distributional reinforcement learning. Because of the inherent non-linearity in the modeling operator, traditional generative models such as energy-based models (EBMs) and score-based generative models (SGMs) cannot be applied to RL contexts; they cannot preserve the linearity of the Bellman update. The proposed new model addresses this challenge by modeling both gradient and scalar fields directly in the distribution space, thereby maintaining linearity and enabling effective generative modeling in MDPs.

The paper introduces divergence-based training methods to optimize neural networks to approximate both fields and defines a specialized SDE for sampling. On the theoretical side, the paper proves that the proposed model will converge to a target distribution (in terms of both KL divergence and Wasserstein distance) regardless of the initial distribution. Experimentally, Bellman Diffusion demonstrates superior performance in estimating and generating target distributions. Also, the proposed model performs well on two OpenAI Gym environments and converges faster than the baseline model.

**Strengths:**

1. The proposed model addresses a key gap in using deep generative models for Markov decision processes. It is well motivated by maintaining the linearity of the Bellman equation.
2. Following the motivation, the authors build a solid theoretical framework around the proposed model. Some important theorems are stated and proved, including the steady-state convergence theorem (Theorem 4.1) and error bounds for neural network approximation (Theorem 4.2). Note that I do not fully follow the proof in the appendix, and cannot guarantee its correctness.
3. The experiments performed are in a variety of domains, including synthetic point distributions, images, and RL enviroments. I especially like the experiments on OpenAI Gym, which demonstrates the fast and stable convergence of the proposed model over the conventional baseline.

**Weaknesses:**

While this paper is primarily focused on methodology and theoretical contributions, I believe there is room for improvement on the experimental side:

1. The abstract claims that the proposed model is a "capable image generator." However, the only image generation results provided are on MNIST (Figure 7, Appendix), and these are purely qualitative. This claim would be better supported with quantitative experiments on real-world image datasets.

2. Are there specific reasons preventing the application of Bellman Diffusion to larger or more complex RL environments? Currently, only two simple examples are shown. Additionally, if possible at all, it would be valuable to compare the performance of Bellman Diffusion with that of denoising diffusion models on these RL tasks, to observe how the non-linearity of traditional diffusion models impacts their performance in this context.

**Questions:**

1. How strong are the assumptions (Assumptions C.1 and C.2) required for proving Theorems 4.1 and 4.2, compared with those typically used in related theoretical analysis? It would be helpful to see more justification regarding their validity. For example, could you provide examples of other works in the field that rely on similar assumptions?

---

> ### Author Response · Authors · 2024-11-24
> **Rebuttal**
>
> We thank the reviewer for such comprehensive and constructive feedback.
>
> > ### [***Weakness-1***] The abstract claims the model is a "capable image generator," but only provides qualitative MNIST results.
>
> We thank the reviewer for highlighting this point. The main goal of this work is to demonstrate the potential of Bellman Diffusion in two aspects:
> - as a method for solving distributional RL tasks;
> - as a proof-of-concept for generative modeling.
>
> In Appendix D.1, we discussed the bottlenecks of applying off-the-shelf deep generative models, including diffusion models (which excel in complex image generation tasks), to learn the value distribution effectively. This highlights the unique strengths of Bellman Diffusion in such contexts.
>
> We follow the convention experiments as in Sliced Score Matching to show Bellman Diffusion is capable of high dimensional data generation, and presented the results in Table 1 (10-20 dimensional space) and Appendix F.1, as a proof-of-concept experiment to show that Bellman Diffusion is not only scalable in terms of data dimensionality but can play a role for pure data generation. Our intention here was not to claim superiority in image generation but to demonstrate that Bellman Diffusion, despite being tailored for RL tasks, can also function as a generative model. We have revised the abstract to clarify this distinction and temper the claim.
>
> > ### [***Weakness-2***] The application of Bellman Diffusion to larger or more complex RL environments? Compare with current denoising diffusion models on these RL tasks?
>
> We thank the reviewer for this insightful comment. In principle, Bellman Diffusion is designed to be a generic framework, and we do not foresee any inherent limitations to its application in more complex RL environments. Based on our understanding, the method’s superior stability and faster convergence should extend to these scenarios, as it learns a continuous return distribution that inherently avoids the discretization errors in the C51 baseline, which accumulate and propagate through state transitions.
>
> To address this, we are currently working to implement Bellman Diffusion in Atari environments from OpenAI’s Gym library, which offer more intricate dynamics and state spaces than the examples provided in the manuscript. While we are striving to complete these experiments during the discussion period, it takes time to adapt our codebase to a new environment (e.g., interaction interface and reward shaping) and the limited timeframe may not allow us to finalize the results. If not included in the rebuttal, we commit to providing these results in the final manuscript to further demonstrate the method’s scalability and applicability to more challenging RL tasks.
>
> > ### [***Question-1***] How do Assumptions C.1 and C.2 compare to typical assumptions in related analyses? It would help to see more justification, including examples from other works.
>
> We appreciate your thoughtful question regarding the strength and validity of Assumptions C.1 and C.2. To address your concern, we would like to highlight that the assumptions made in our manuscript are standard and widely used in theoretical analysis within the context of diffusion models and Langevin dynamics.
>
> Specifically, Assumptions C.1 and C.2 relate to:
> 1. Smoothness and Lipschitzness of the target distribution $p_{\mathrm{target}}$,
> 2. Its fast decay at infinity, and
> 3. The Log-Sobolev inequality.
>
> These are commonly encountered in the literature, and we believe they are both reasonable and necessary for ensuring the convergence properties that we derive.
>
> For example:
> - In [1, 2, 3], the authors assume certain  smoothness and Lipschitz continuity of the density functions, which is essential for establishing well-behaved gradients and ensuring convergence.
> - In [2] and [4], the Log-Sobolev inequality is assumed, which is crucial for controling the entropy of a density with respect to its gradient.
> - Additionally, [1, 3] make assumptions on the decay of the density at infinity to control tail behaviors and guarantee that the diffusion process remains well-behaved in the limit.
>
>
> We hope this provides the clarification you requested, and we would be happy to discuss further or provide additional references if needed.
>
> **References**
>
> [1] Maximum likelihood training of score-based diffusion models. Advances in neural information processing systems, 34, 1415-1428.
>
> [2] Convergence for score-based generative modeling with polynomial complexity. Advances in Neural Information Processing Systems, 35, 22870-22882.
>
> [3] Maximum likelihood training for score-based diffusion odes by high order denoising score matching. In International Conference on Machine Learning (pp. 14429-14460). PMLR.
>
> [4] Rapid convergence of the unadjusted langevin algorithm: Isoperimetry suffices. Advances in neural information processing systems, 32.

---

> > ### Comment · Reviewer_sSvE · 2024-11-30
> >
> > Thank you for your reply! I appreciate the clarification from the authors and will keep my score which is above the accepting threshold.

---

> > > ### Author Response · Authors · 2024-12-01
> > > **Thank you**
> > >
> > > We sincerely appreciate the reviewers' feedback in helping us improve this work. If there are any further questions requiring clarification, we would be happy to address them.

---

### Official Review · Reviewer_SqjE · 2024-11-10

**Soundness:** 1
**Presentation:** 1
**Contribution:** 2
**Rating:** 3
**Confidence:** 3

**Summary:**

The authors propose a new genarative model, Bellman diffusion, that only depends on the density and its derivative. This choice enables learning return distributions in distributional RL as it's consistent with the linearity requirement of an MDP.

**Strengths:**

* The author propose a sampling technique that combines both the target density and its derivative and show that it theory it leads to convergence to the target.

**Weaknesses:**

* Structure and clarity: The paper can be restructured in a better way.  For instance, the motivation would be clearer if it would have integrated some material from Sec. 21 can help convey in the intro. Also, in the intro, the authors refer to equations (83) from the approach section. This assumes that the paper has to be read twice?

* Notation issues:
** x is undefined in Eq1
** The prime is Eq2, line 2, right most term, should be applied to z
** Extra dot in line 185

* Formulation lacks preciseness:
** Eq 1 in linear in p but not linear in x. The fact that the linearity is related to the operator can only be understood in Sec 2.2
** The explanation of eq2 can be clearer is it would have explicitly mentioned that the energy formulation transforms the equality in eq2 into an inequality.

* Approach:
** What's the use of the isotropic Gaussan in Eq 199? There are no references to how this proxy is derived. It's similar to score matching with a new Gaussian term?
** The score matching loss is not scalable (even the sliced version is not). It's established that score matching is not equivalent to fisher divergence because the derivation requires integration by parts which assumes access to an infinite number of samples. This is not the case in practice.
** A sub-section or a figure/algorithm showcasing how this generative model can be used in distributional RL is missing.

**Questions:**

* Please, check weakness section.
* Instead of Eq12, why not use SVGD sampler and the derived distribution following:
Messaoud S, Mokeddem B, Xue Z, Pang L, An B, Chen H, Chawla S. S $^ 2$ AC: Energy-Based Reinforcement Learning with Stein Soft Actor Critic. ICLR., 2024 ?
The sampler doesn’t depend on the score of the updated particle itself, which makes it have the desired property of linearity.

---

> ### Author Response · Authors · 2024-11-24
> **Rebuttal, Section 1**
>
> We thank the reviewer for such comprehensive and constructive feedback.
>
> > ### [***Weakness-1,2,3***] Structure and clarity, notation issues, and formulation lacking preciseness
>
>  We sincerely thank the reviewer for their insightful feedback, which has helped us identify areas for improving the clarity, presentation, and precision of our manuscript. Below is a summary of the changes we have made in response to your comments (marked in blue in the revised manuscript):
>
> **[Clarity and Presentation in Introduction]**
> 1. Revised the introduction to emphasize MDP and distributional RL tasks.
> 2. Elaborated on the nonlinearity in DGM modeling and the desired linearity in the Bellman equation, which relates a state’s return distribution linearly to those of future states—reinforcing our motivation.
> 3. Provided a detailed and rigorous explanation of Energy-Based Models (EBMs) as in Sec. 2.
> 4. Removed confusing cross-references to improve overall clarity.
> 5. Improved with figure quality, with better readability.
> 6. Added an experiment to show that Bellman Diffusion is better than common diffusion models in recognizing the data imbalance.
>
> **[Notations and Formulas]**
>
> Revised notations and formulas to ensure all elements are explicitly defined and consistent throughout the manuscript.
>
> **[Elaboration on Applying Bellman Diffusion to Distributional RL]**
>
> To further elaborate on how Bellman Diffusion is applied to MDPs, particularly in distributional RL tasks, we have moved and elaborated the contents originally in Appendix D.2 to the current Section 5.
>
> Thank you again for the valuable comments and careful check. If there are additional unclear points, please kindly let us know, and we will be happy to incorporate your advice to further improve our paper.
>
> > ### [***Weakness-4***] Approach: Eq 199's isotropic Gaussian is unclear, resembling score matching. Score matching is not scalable and differs from Fisher divergence. The generative model's application in distributional RL is missing.
>
> **[About Isotropic Gaussian in Proposition 3.1]**
>
> We appreciate the reviewer’s insightful comments. The use of the isotropic Gaussian in Line 199 is not strictly necessary; it serves as a convenient choice for constructing a family of distributions $\\{q_{\epsilon} \\}_{\epsilon>0}$ that approximates the Dirac delta distribution as $\epsilon \to 0$ to derive a feasible and simple training objective.
>
> To address the reviewer’s concerns, we have expanded the motivation and provided a more detailed explanation in the revised manuscript, specifically highlighting the rationale for using the isotropic Gaussian in deriving Proposition 3.1. Below, we outline the derivation of Proposition 3.1 using $L_{grad}(\phi)$ as an example. A similar argument applies to $L_{id}(\varphi)$. First, we recall the definition of $L_{grad}(\phi)$ is
> $$
> L_{grad}(\phi) = D_{grad}({p_{target}(\cdot)}, g_{\phi}(\cdot)) = E_{x} \Big[ \| \nabla {p_{target}(x)} - g_{\phi}(x)  \|^2 \Big].
> $$
> As the reviewer noted, integration by parts and the divergence theorem simplify it to:
> $$
>    L_{grad}(\phi) = C_{grad} + \int  p_{target}(x) \| g_{\phi}(x) \|^2 dx + \int p_{target}(x)^2 tr( \nabla g_{\phi}(x) ) dx,
> $$
> where $C_{\mathrm{grad}}$ is constant with respect to $\phi$. The key challenge lies in handling the squared probability term $p_{\mathrm{target}}(x)^2$.
>
> To address this, we aim to make the integral
>  $\int p_{target}(x)^2 tr( \nabla g_{\phi}(x) ) dx$ estimable via Monte Carlo sampling. More generally, we seek to reduce the following term $\mathcal{I}$:
> $$
> \mathcal{I} = \int p_{target}(x)^2 f(x) dx.
> $$
>     to a form suitable for Monte Carlo estimation. Here, $f(x)$ is a general scalar-valued function. We denote the Dirac delta distribution as  $\delta(\cdot)$, where $\delta(y - x)=1$ if $y=x$; otherwise, $\delta(y - x)=0$.  Using the decoupling trick from standard probability theory argument:
> $$
> \mathcal{I} = \int p_{arget}(x) f(x) p_{target}(x) dx  = \int_{x} p_{target}(x) f(x) ( \int_{y} p_{arget}(y) \delta(y - x) dy ) d x
>            = \int_{x} \int_{y} p_{target}(x)p_{target}(y) f(x) \delta(y - x) dx dy,
> $$
> which can be further simplified as
> $$
> \mathcal{I} = E_{x_1, x_2 \sim p_{arget}}[f(x_1) \delta(x_2 - x_1)].
> $$
> For continuous data, the Dirac delta output often vanishes, so we approximate it with a sequence of Gaussian distributions (in the distributional sense): $N(x_2 - x_1; 0, \epsilon I_D) \rightarrow \delta(x_2 - x_1)$ as $\epsilon \rightarrow 0^+$. This gives:
> $$
> \mathcal{I} = E_{x_1, x_2 \sim p(x)}[f(x_1) \delta(x_2 -x_1)] = E_{x_1, x_2}[f(x_1) \lim_{\epsilon \rightarrow 0} N(x_2 - x_1, 0, \epsilon I_D)] =  \lim_{\epsilon \rightarrow 0} E_{x_1, x_2 \sim p(x)} [f(x_1)  N(x_2 - x_1, 0, \epsilon I_D)].
> $$
> Thus, we derive the practical training objective:
> $$
>     L_{grad}(\phi) = C_{grad} + \lim_{\epsilon \rightarrow 0} E [ \| g_{\phi}(x_1) \|^2 + tr( \nabla g_{\phi}(x_1))  N(x_2 - x_1; 0, \epsilon I_D ) ].
> $$

---

> ### Author Response · Authors · 2024-11-24
> **Rebuttal, Section 2**
>
> In the discrete setting, this problem is much simpler:  $\mathcal{I} = \sum_{x} p_{target}^2(x) f(x)$. The term $\mathcal{I}$ can be converted into
> $$
> \mathcal{I} = \sum_{x, y} p_{target}(x)p_{target}(y) f(x) \delta(y - x) = E_{x_1, x_2 \sim p_{target}}[  f(x_1) \delta (x_2 - x_1)].
> $$
> Here, no approximation is required, as $\delta(x_2 - x_1)$ corresponds to a one-hot vector for discrete data.
>
> **[About Sliced (Score) Matching]**
> We appreciate the reviewer’s insightful comments.
>
> -  We would like to clarify that our primary focus is on distributional RL, where we model the distribution of returns (a 1-dimensional quantity), so scalability is not a critical issue in this context. While scaling to higher-dimensional settings is not the main focus of our work, we did include experiments with higher-dimensional data (please kindly see Table 1 and Appendix F.1) to demonstrate the generalizability of our method. As a future direction, we propose two possible approaches to improve scalability for higher-dimensional tasks: (1) employing as diffusion model's manner that gradually adds multi-level noise to avoid the slice trick, and (2) exploring improved configurations for the latent space in Bellman Diffusion.
> - Regarding the assumption of smooth data density (infinite data) in the integration by parts of our objective function derivation, we argue that these assumptions are not prohibitive in practical distributional RL scenarios, where the model can interact with the environment infinite times.
>
> In principle, we may follow an argument similar to Theorem 2 of the Sliced Score Matching paper, suggesting that the estimator obtained by replacing the expectation over $p_{target}$ with Monte Carlo sampling (sample size $N$) in Proposition 3.1 is a *consistent estimator*. As the sample size $N$ increases, the estimator becomes progressively closer to the true optimizer of $L_{grad}$ (or $L_{id}$). In practice, especially in RL, the ability to sample arbitrarily many data points improves estimator accuracy, addressing the concern related to integration by parts.
>
> If the reviewer suggests including a more formal proof to support this argument, we would be happy to expand on this in our manuscript.
>
> **[About Elaboration on Distributional RL]**
> We kindly refer the reviewer to Appendix D.2 and Algorithm 3 in the original manuscript, where we outline the application of Bellman Diffusion to general planning tasks, including distributional RL. To enhance clarity and visibility, we have revised the main text by moving and elaborating the algorithmic details of Bellman Diffusion's application to distributional RL, previously in Appendix D.2, into Section 5 of the revised manuscript.
>
> > ### [***Question-2***] SVGD sampler and new reference: $S^2AC$
>
> We thank the reviewer for highlighting this reference. While the S$^2$AC framework and its use of the SVGD sampler are indeed valuable contributions, they do not align with the problem setting and objectives of our paper. Specifically:
>
> **Different Targets** S$^2$AC is designed for Maximum Entropy (MaxEnt) RL, where the primary goal is learning a stochastic policy. In contrast, our work focuses on distributional RL, which aims to model the return distribution for each state. These are fundamentally different RL paradigms, addressing distinct challenges and requiring tailored methodologies.
>
> **Score function dependency** Similar to the Langevin dynamics used in vanilla diffusion models, the SVGD sampler relies on the score function. This dependency introduces challenges when applied to distributional RL, where we aim to model the return distribution without direct reliance on the score of the updated particle. For more details on these considerations, please refer to Sec. 2 of our paper.
>
> While we acknowledge the strengths of S$^2$AC and SVGD in their respective domains, their approach is not directly applicable to the distributional RL setting we address. We appreciate the reviewer’s suggestion and have clarified this distinction in the manuscript (please see Appendix A of the revised manuscript).

---

> ### Comment · Reviewer_SqjE · 2024-11-25
>
> What about computing the expectation in S2AC over particles different from the updated one. This way you can get rid of the score term dependency on the particle. Also, you can choose the kernel to be the bilinear kernel. The svgd transformation becomes linear. I am worried that grad p and p are too small quantities that would result in inefficient sampling.

---

> > ### Author Response · Authors · 2024-11-27
> >
> > We appreciate your comments and the opportunity to provide further clarification.
> > - For your further concern about S2AC, we have not figured out a concrete way following your suggestions, which can cross out the score function in SVGD. In particular, the number of particles is inherently finite, so it is less likely that one can reduce the score term in theory through empirical expectation over finite particles. A bilinear kernel will also not change its inherent position in SVGD, which cannot make the score term linear. To make things clear, could you please provide strict mathematics regarding your point?
> > - For your concern about inefficient sampling, please refer to the experiment results (e.g., Fig. 4 and Fig. 5) in our revised paper, showing that our sampling dynamics can generate high-quality samples.

---

### Meta-Review · Area_Chair_mvpU · 2024-12-29

**Metareview:**

This paper introduces Bellman Diffusion, which exploits advanced generative models for distributional reinforcement learning with linearity in MDPs. The method is well-motivated to fill the gap in distributional RL, and derived with the practical objectives.

However, the major issues raised by the authors lies in the following aspects:

1, The presentation of the methodology is difficult to follow.

2, The empirical evaluation is mainly focusing on simple RL environments, with limited baselines. Therefore, it is difficult to justify the advantages in practice.

In sum, the paper proposed an interesting method for distributional RL, however, it is not ready enough to be published yet.

**Additional Comments On Reviewer Discussion:**

The authors did not fully address the major issues raised by the reviewers. Therefore, the major reviewers kept their opinions.

---

### Decision · Program_Chairs · 2025-01-22

Reject